EMBO
*reports*

# Non-apoptotic caspase activation preserves *Drosophila* intestinal progenitor cells in quiescence

Lewis Arthurton[1], Dominik Antoni Nahotko[2,†], Jana Alonso[3,†] (iD), Franz Wendler[1] &
Luis Alberto Baena-Lopez[1,*] (iD)

## Abstract

Caspase malfunction in stem cells often precedes the appearance and progression of multiple types of cancer, including human colorectal cancer. However, the caspase-dependent regulation of intestinal stem cell properties remains poorly understood. Here, we demonstrate that *Dronc*, the *Drosophila* ortholog of *caspase-9/2* in mammals, limits the number of intestinal progenitor cells and their entry into the enterocyte differentiation programme. Strikingly, these unexpected roles for Dronc are non-apoptotic and have been uncovered under experimental conditions without epithelial replenishment. Supporting the non-apoptotic nature of these functions, we show that they require the enzymatic activity of Dronc, but are largely independent of the apoptotic pathway. Alternatively, our genetic and functional data suggest that they are linked to the caspase-mediated regulation of Notch signalling. Our findings provide novel insights into the non-apoptotic, caspase-dependent modulation of stem cell properties that could improve our understanding of the origin of intestinal malignancies.

**Keywords** Dronc; *Drosophila* caspases; intestinal homeostasis; intestinal progenitor cells; non-apoptotic caspase functions
**Subject Categories** Autophagy & Cell Death; Development; Stem Cells & Regenerative Medicine

## Introduction

Caspases are cysteine-dependent aspartate-specific proteases responsible for the implementation of apoptosis (Xu *et al*, 2018). Recently, an emerging body of evidence is also attributing non-apoptotic functions to these enzymes in a wide variety of cell types (Arama *et al*, 2020). In particular, sublethal caspase activity has been shown to modulate the proliferation and differentiation of stem cells through mechanism which is still largely unknown (Bell & Megeney, 2017; Baena-Lopez *et al*, 2018b). Previous studies have

indicated that *caspase*-9 deficiency in patients suffering from colorectal cancer appears to stimulate the proliferation of intestinal precursors, whilst compromising their differentiation (Xu *et al*, 2013). Furthermore, our laboratory has recently uncovered stereotyped patterns of non-apoptotic caspase activation in *Drosophila* progenitor cells (Baena-Lopez *et al*, 2018a). Therefore, we decided to investigate the origin and biological significance of such caspase activation patterns in the *Drosophila* intestinal system.

The evolutionary conservation of gene function and the ease of gene manipulation in *Drosophila melanogaster* have been routinely exploited to uncover many genetic networks and cellular processes connected with human diseases (Miles *et al*, 2011). Accordingly, important discoveries regarding intestinal stem cell biology and caspases have been obtained using fruit flies (Miguel-Aliaga *et al*, 2018). The caspases are expressed as pro-enzymes that only become fully active after one or more steps of proteolytic processing (Goyal *et al*, 2000; Srinivasula *et al*, 2002; Crawford & Wells, 2011; Parrish *et al*, 2013; Xu *et al*, 2018; Baena-Lopez *et al*, 2018b). In *Drosophila*, the intrinsic apoptotic programme is initiated by the upregulation of different pro-apoptotic proteins (Hid, Reaper, Grim and/or Sickle) (Goyal *et al*, 2000; Christich *et al*, 2002; Srinivasula *et al*, 2002; Wing *et al*, 2002). These proteins counteract the activity on caspases of the *Drosophila* inhibitors of apoptosis (DIAP-1 and DIAP-2) (Steller, 2008; Leulier *et al*, 2006; Huh *et al*, 2007; Silke & Meier, 2013). In pro-apoptotic conditions, the main *Drosophila* initiator caspase referred to as Dronc (caspase-2/9 orthologue in mammals) can molecularly interact with Dark-1 (Apaf-1), forming a protein complex termed the apoptosome. These events elicit high levels of activation of *Dronc* (Rodriguez *et al*, 1999; Muro *et al*, 2004; Xu *et al*, 2005; Cheng *et al*, 2017), and the subsequent enzymatic processing of the effector caspases (*Death caspase-1, DCP-1 (caspase-7)*; the *death-related ICE-like caspase, drICE (caspase-3)*; *Death-associated molecule related to Mch2 caspase, Damm;* and the *Death executioner caspase related to Apopain/Yama, Decay*). Upon activation, effector caspases disrupt all of the essential subcellular structures leading to cell death (Parrish *et al*, 2013; Baena-Lopez *et al*, 2018b).

The *Drosophila* intestine contains a subset of intestinal stem cells (ISCs), responsible for the renewal of the gut epithelium (Micchelli & Perrimon, 2006; Ohlstein & Spradling, 2006; Jiang & Edgar, 2011).

1 Sir William Dunn School of Pathology, University of Oxford, Oxfordshire, UK
2 Northwestern University Feinberg School of Medicine, Chicago, IL, USA
3 Laboratorio de Agrobiología Juan José Bravo Rodríguez (Cabildo Insular de La Palma), Unidad Técnica del IPNA-CSIC, Santa Cruz de La Palma, Spain
*Corresponding author. Tel: +44 (0) 1865618653; E-mail: alberto.baenalopez@path.ox.ac.uk
†These authors contributed equally to this work

ISCs upon demand can also differentiate as either intermediate progenitor cells termed enteroblasts (EBs), or fully differentiated secretory cells called enteroendocrine cells (EEs) (Li & Jasper, 2016). The EBs rarely, if ever, divide but can terminally differentiate as mature absorptive cells referred to as enterocytes (ECs) (Zhai *et al*, 2017). High levels of the Notch pathway components, *Delta* (*Dl*) and *Supressor of Hairless* (*Su(H)*), are considered cell identity markers of ISCs and EBs, respectively. In addition, these cell types also express the stem cell marker *escargot* (*esg*) (Micchelli & Perrimon, 2006). Loss of *esg* expression and upregulation of *nubbin* (*nub*, hereafter Pdm1) facilitate the conversion of EBs into ECs (Tang *et al*, 2018). The expression of Prospero is the distinctive marker of EE cells that directly derive from ISCs (Lemaitre & Miguel-Aliaga, 2013; Guo & Ohlstein, 2015; Zeng & Hou, 2015). An abundant body of literature has described the genetic factors controlling the proliferation and differentiation of ISCs into EBs; however, the conversion of EBs into ECs is less characterised, particularly in non-regenerative conditions without tissue replenishment. The *Notch* pathway is one of the critical signalling cascades that regulates gut epithelial homeostasis (Zwick *et al*, 2019). Low levels of Notch signalling promote the self-renewal of ISCs, whilst elevated Notch activation stimulates their transition into EBs (Hakim *et al*, 2010; Guo & Ohlstein, 2015; Zwick *et al*, 2019). High levels of Notch signalling combined with specific transcription factors also promote the entry of EBs into the EC differentiation programme (Kapuria *et al*, 2012; Guo & Ohlstein, 2015). In tissue damaging conditions, Notch activation can modulate additional signalling pathways, such as JAK-STAT and BMP signalling, that stimulate EB differentiation (Zhai *et al*, 2017). Interestingly, caspase malfunction has also been shown to alter the proliferation and differentiation of ISCs in regenerative conditions (Jin *et al*, 2013). Here, we describe how the non-apoptotic activity of the initiator *Drosophila* caspase Dronc limits the number of progenitor cells and their entry into the EC differentiation pathway, under experimental conditions without basal epithelial replenishment. Importantly, these novel non-apoptotic caspase functions are strongly linked to the caspase-dependent modulation of Notch signalling.

## Results

### Adult intestines reared in experimental conditions without homeostatic cellular turnover show a stereotyped non-apoptotic caspase activation pattern

The laboratory developed a caspase sensor to specifically detect the activity of initiator caspases in *Drosophila* tissues (DBS-S-QF sensor) (Baena-Lopez *et al*, 2018a). This sensor relies on a truncated form of *Drice,* attached to the cell membranes at the N terminus. The C terminus contains the transcriptional activator QF (Baena-Lopez *et al*, 2018a), which upon initiator caspase-mediated cleavage, is released and is translocated to the nucleus. The presence of QF in the nucleus can induce the transcriptional activation of any cDNA of interest downstream of the QUAS regulatory sequences (Baena-Lopez *et al*, 2018a). Activating cellular markers with variable lifetimes with this tool, we uncovered a stereotyped pattern of non-apoptotic caspase activation in the progenitor cells of the posterior midgut (R4/5) of adult *Drosophila* females (Fig 1A) (Baena-Lopez

*et al*, 2018a). Similar sensors for effector caspases have previously shown reproducible and preferential non-apoptotic activation in ECs (Appendix Fig S1) (Ding *et al*, 2016). Following these initial observations, we sought to assess the impact of these caspase activation patterns on intestinal homeostasis. To evaluate the intestinal epithelial status in our experimental conditions, we took advantage of the ReDDM cell lineage-tracing system (Antonello *et al*, 2015). This method allows the investigation of intestinal epithelial turnover, through comparing the expression of a short-lived green fluorescent protein (GFP) and a highly stable Histone protein tagged with a red fluorescent molecule (hereafter Histone-RFP) (Antonello *et al*, 2015). This lineage-tracing tool, under the regulation of the *esg*-Gal4 driver, labels undifferentiated intestinal progenitor cells with both fluorescent markers (ISCs and EBs; Fig 1B) (Antonello *et al*, 2015). However, the silencing of the *esg* promoter during EB differentiation results in the rapid degradation of the GFP signal, whilst the Histone-RFP marker persists; the visualisation of Histone-RFP without GFP can be used as a proxy of differentiation (Antonello *et al*, 2015). This cell lineage-tracing method failed to show signs of intestinal regeneration in our experimental conditions (Fig EV1A) during the first 7 days post-ReDDM induction (Fig 1B and C), and both fluorescent signals overlapped in the intestinal progenitor cells (*esg*-positive cells; Fig 1B and C) (Antonello *et al*, 2015). Despite not observing signs of regeneration, caspases were intriguingly activated in a characteristic pattern under such experimental conditions (DBS-S-QF; Fig 1A, Baena-Lopez *et al*, 2018a). Expectedly, unspecific tissue damage triggered by either exposure to paraquat [reactive oxygen species (Ali *et al*, 1996)] or detrimental dietary conditions, caused robust caspase activation (Figs 1D and EV1B) and epithelial regeneration (Figs 1E and F, and EV1C and D); note the presence of newly differentiated ECs with a large nuclear size expressing Histone-RFP alone (compare Fig 1B with E). These results suggested that our experimental conditions preserve the intestinal epithelia in quiescence state, without cellular death during the first 7-day post-ReDDM activation. Supporting this conclusion, the overexpression of the effector caspase inhibitor P35 (Hay *et al*, 1994) did not cause morphological or cellular alterations in the intestinal epithelium (Fig 1G–J). Collectively, these data indicated the non-apoptotic nature of DBS-S-QF patterns under our experimental regime (Baena-Lopez *et al*, 2018a), whilst prompting questions regarding its biological significance.

### The initiator caspase Dronc prevents unintended differentiation of intestinal progenitor cells

Since P35 only prevents the activity of effector caspases, we investigated the potential implication of initiator caspases in the intestine. To this end, genome engineering protocols (Baena-Lopez *et al*, 2013) allowed us to create a new *Dronc* knockout (*Dronc*[KO]) allele that contains an attP-integration site immediately after the *Dronc* promoter (Fig EV2A and B). As with the original *Dronc* null alleles (Chew *et al*, 2004; Xu *et al*, 2005), our *Dronc*[KO] mutant was lethal during early pupal development, either in homozygosis or in combination of other amorphic alleles (Fig EV2C). Importantly, the heterozygous insertion of a wild-type *Dronc* cDNA into the *Dronc* attP site gave rise to fertile adult flies, similar to their wild-type siblings (Fig EV2D–F). These results indicated that our rescue construct retained all of the essential

functionality of the endogenous gene, whilst validating the attP-integration site. Next, we created a conditional allele of *Dronc* (*Dronc*[KO-FRT Dronc-GFP-Apex FRT QF]; Fig EV2G) that included a *Dronc* cDNA flanked by FRT recombination motifs and it was able to rescue the lethality of our *Dronc*[KO] mutant. Additionally, this allele had a QF transcriptional activator downstream of the FRT-rescue-cassette (*Dronc*[KO-FRT Dronc-GFP-Apex FRT QF]). Upon *flippase*-mediated excision of the FRT cassette, the production of QF under the transcriptional regulation of *Dronc* enabled the expression of any QUAS transgene (e.g. QUAS-LacZ). This feature can be used to identify excision events in *Dronc*-transcribing cells and modulate their gene expression profile. Proving the efficiency of our method, 3 days after *flippase* induction, 91.35% of *esg*-labelled cells (GFP-positive cells) showed the transcriptional activation of the *lacZ* reporter gene (Appendix Fig S2A and B). Additionally, we noticed signs of hyperplasia and enlarged *LacZ*-positive cells without *esg* expression (GFP-negative cells) (Appendix Fig S2A). Comparable cellular alterations were observed using a different conditional allele that expressed a suntag-HA-Cherry chimeric protein (*Dronc*[KO-FRT Dronc-GFP-Apex FRT Suntag-HA-Cherry]) (Figs 2A–E and EV2H), thus excluding the association of our original phenotypes with potential QF toxicity (Riabinina & Potter, 2016). Beyond intestinal hyperplasia (Fig 2C and Appendix Fig S2C), the allele conversion using the suntag-HA-Cherry construct caused cellular enlargement (Fig 2D and Appendix Fig S2D), and EB differentiation as ECs; note the co-expression of Histone-RFP with the EC maker Pdm-1, and the co-localisation between the GFP and Pdm1 (inset in Fig 2B and E, Appendix Fig S2E–G). Furthermore, these phenotypes worsened over time (Fig 2C–E). Discarding any detrimental effects linked to the expression of suntag-HA-cherry, flies expressing a WT version of Dronc tagged with suntag-HA-Cherry (*Dronc*[KO-FRT Dronc-GFP-Apex FRT-FLWT-Suntag-HA-Cherry]) did not display any noticeable intestinal phenotypes (Fig EV3A and F–H, and Appendix Fig S2H). These results were further confirmed inducing loss-of-function genetic mosaics of a well-characterised *Dronc* null allele (*Dronc*[I29]) (Appendix Fig S2I–K); notice the increased number of cells forming *Dronc*[I29] mitotic recombination clones and the presence of large differentiated cells. Importantly, all of these observations were collected in experimental conditions without tissue replenishment and cell death; therefore, we conclude that non-apoptotic activation of Dronc contributes to preserving the intestinal progenitor cells in quiescence (non-proliferative and undifferentiated). To further characterise the differentiation status of *Dronc* mutant intestines, we next analysed the expression profile of metabolic enzymes which are highly enriched in ECs (Doupe et al, 2018). Interestingly, several of these genes were transcriptionally downregulated in our mutant conditions, whilst others were upregulated (Appendix Fig S2L–N). This abnormal gene expression profile of EC markers could be indicative of either premature or partial differentiation of *Dronc* mutant progenitor cells.

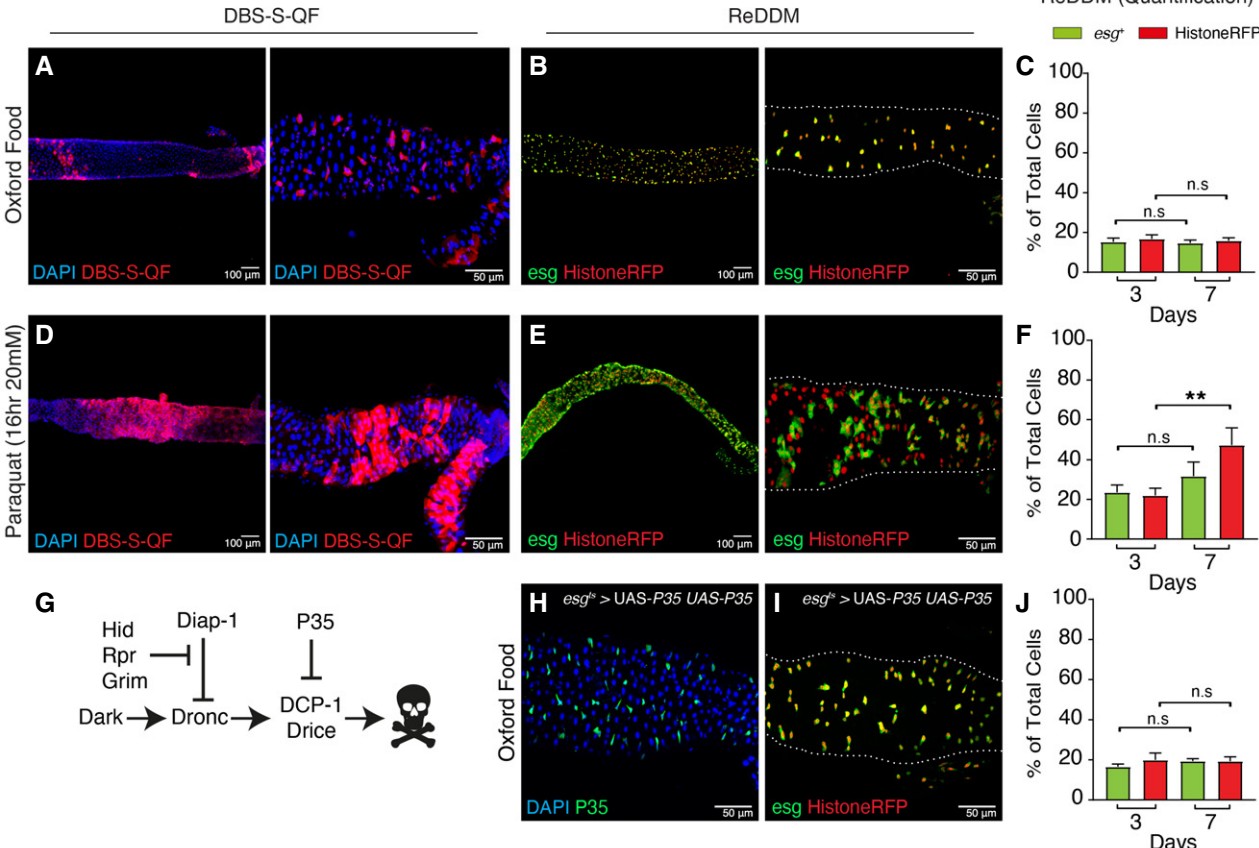

**Figure 1.**

**Figure 1.  Non-apoptotic caspase activation in the Drosophila Intestinal System without the need for tissue replenishment.**

A   A representative example of a 7 days (7d) old adult female posterior midgut at low (A) and high magnification (B, Region 4-Region 5) showing the initiator caspase reporter DBS-S-QF (Red, immunostaining anti-HA). Flies were reared under experimental regime that protects the intestinal epithelial integrity (intestine reared in Oxford Medium and flies transferred every 2 days into vials with fresh food, see Fig EV1) at 25°C. The image on the left was acquired at low magnification with the 10× objective, whilst the right panel was acquired using the 40×. The left and right images in all examples correspond to independent intestines of the same genotype and treatment. Genotype: $y^1\ w^{1118}$ UAS-mCD8::GFP.L QUAS-mtdTomato-3xHA Act-DBS-S-QF (BL83131).

B   A representative example of a 7-d intestine reared at 29°C in experimental conditions that protect the epithelial integrity and labelled with ReDDM cell lineage tool. *esg* (green cytoplasmic signal) labels intestinal progenitor cells, whilst Histone-RFP (red nuclear signal) acts as a semi-permanent marker that allows the visualisation of differentiated cells. Note the extensive overlap between the two markers and the absence of differentiated cells only showing the Histone-RFP labelling, as an indication of negligible epithelial turnover. The image on the left was acquired at low magnification with the 10× objective, whilst the right panel was acquired using the 40×. White dotted line outlines the gut using DAPI staining (not shown) as a reference. Genotype: $w^{1118}$; *esg*-Gal4 UAS-CD8-GFP (Irene Miguel Aliaga)/Cyo; UAS-Histone-RFP (BL7019) TubG80ts (BL7019).

C   Quantification of intestinal cell subpopulations labelled with ReDDM system at high magnification (GFP and Histone-RFP) and different time points post-ReDDM activation (3 and 7 days) in experimental conditions that protect the epithelial integrity; note that none of the cell populations in the gut (GFP (n.s. no significant, $P = 0.5267$) or Histone-RFP (n.s., $P = 0.2752$) significantly increase in number overtime (Quantifications were made using $N \geq 2$ biological replicates, total number of guts analysed at 3 days $n = 34$ and at 7 days $n = 45$; Mann–Whitney, C). Error bars represent standard error of the mean. Genotype: $w^{1118}$; *esg*-Gal4 UAS-CD8-GFP (Irene Miguel Aliaga)/ Cyo; UAS-Histone-RFP (BL7019) TubG80ts (BL7019).

D   A representative example of an adult female posterior midgut at low (A) and high magnification (B, Region 4-Region 5) showing the initiator caspase reporter DBS-S-QF (Red, immunostaining anti-HA) after 16 h of paraquat treatment in Oxford food at 25°C; note the expansion of the labelling with DBS-S-QF to large intestinal cells (ECs) (compare D with A). The image on the left was acquired at low magnification with the 10× objective, whilst the right panel was acquired using the 40×. White dotted line outlines the gut using DAPI staining as a reference. Genotype: $y^1\ w^{1118}$ UAS-mCD8::GFP.L QUAS-mtdTomato-3xHA Act-DBS-S-QF (BL83131).

E   ReDDM lineage-tracing in an adult intestine reared in Oxford Medium and paraquat (20 mM) during 16 h at 29°C; note the abundance of Histone-RFP cells without GFP signal, as an indication of epithelial damage and subsequent differentiation of progenitor cells. The image on the left was acquired at low magnification with the 10× objective, whilst the right panel was acquired using the 40×. Genotype: $w^{1118}$; *esg*-Gal4 UAS-CD8-GFP (Irene Miguel Aliaga)/Cyo; UAS-Histone-RFP (BL7019) TubG80ts (BL7019).

F   Quantification of ReDDM labelling at high magnification after paraquat treatment; note the statistically significant increase (**$P = 0.0099$) of Histone-RFP expressing cells without GFP signal (Quantifications were made using $N \geq 2$ biological replicates, 3d $n = 9$, 7d $n = 7$). Error bars represent standard error of the mean. Genotype: $w^{1118}$; *esg*-Gal4 UAS-CD8-GFP (Irene Miguel Aliaga)/Cyo; UAS-Histone-RFP (BL7019) TubG80ts (BL7019).

G   A simplified depiction of the apoptotic pathway in *Drosophila*. The pro-apoptotic factors: Hid, Grim and Reaper, stimulate the formation of the apoptosome (Dark and Dronc), resulting in the activation of the effector caspases (Drice & DCP-1) and cell death. Overexpression of the P35 baculoviral protein inhibits the activity of the effector caspases and cell death.

H   Representative example of an intestine 7d old expressing two copies of the effector caspase inhibitor P35 under the regulation of *esg*-gal4 (green immunostaining with antibody against P35) at 29°C. Genotype: $w^{1118}$; *esg*-Gal4 UAS-CD8-GFP/ UAS-P35 (BL5072); UAS-P35 (BL5073)/+.

I   ReDDM lineage-tracing system in a *Drosophila* intestine expressing two copies of the effector caspase inhibitor P35 under the regulation of *esg*-Gal4 and experimental conditions that protect the epithelial integrity at 29°C. Genotype: $w^{1118}$; *esg*-Gal4 UAS-CD8-GFP/ UAS-P35 (BL5072); UAS-Histone-RFP TubG80ts/ UAS-P35 (BL5073).

J   ReDDM quantification corresponding to the intestines described in (I); no significant increase in either *esg* (n.s., $P = 0.1352$) or Histone-RFP (n.s., $P = 0.9801$) cell number is observed (Quantifications were made using $N \geq 2$ biological replicates; unpaired two-tailed *t*-test, 3d $n = 12$, 7d $n = 11$). Error bars represent standard error of the mean. DAPI (blue) labels the nuclei in panels A, D and H. Genotype: $w^{1118}$; *esg*-Gal4 UAS-CD8-GFP/ UAS-P35 (BL5072); UAS-Histone-RFP TubG80ts/ UAS-P35 (BL5073).

## Dronc functions require its enzymatic activity but are largely independent of the apoptotic cascade

Some of the functions attributed to *Dronc* rely on protein–protein interactions, instead of its enzymatic activity (Ouyang *et al*, 2011). Therefore, we decided to investigate the molecular features of Dronc required in our experimental scenario. To this end, we utilised a new set of conditional alleles that expressed enzymatically inactive versions of *Dronc* upon FRT-rescue cassette excision (Fig EV2J–L; please see details in the Materials and Methods section). All of these alleles replicated the previously described *Dronc*-mutant phenotypes (Fig EV3C–H). Importantly, all of these mutants showed similar expressivity and protein stability, thus discarding any relation of these phenotypes with their differential expression (Fig EV2M). In addition, these findings indicated that the enzymatic activity of Dronc is required for the newly described functions. Next, we explored whether the primary substrates of *Dronc* during apoptosis, the effector caspases (*drIce, Dcp-1, Decay* and *Damm*), were relevant in our experimental context. Their downregulation failed to replicate the proliferation and differentiation phenotypes observed in *Dronc* LOF conditions (Fig EV4A–D). Comparable results were obtained downregulating the expression of Dronc-activating factors, such as Dark and the pro-apoptotic factors Reaper, Hid and Grim

(Fig EV4E–G). These results separate Dronc functions in progenitor cells from the apoptosis programme (see Discussion). They also confirmed the lack of cell death and negligible epithelial turnover in our experimental regime.

## The non-apoptotic function of Dronc is required in ISCs and EBs

Our previous experiments could not ascribe the novel *Dronc* functions to a specific gut progenitor cell type. To address this question, we targeted the expression of *Dronc* in ISCs using the *Dl*-Gal4 driver (Zeng *et al*, 2010). As noted with *esg*-Gal4, the excision efficiency of the FRT-rescue cassette in ISCs using the *Dl*-Gal4 line was also very high (81.15%; Appendix Fig S2O and P). However, *Dronc* deficiency in ISCs did not cause epithelial alterations; no increase of Delta-positive cells or cell size (Fig 3A–D). We could not observe either a differentiation bias towards EE fate quantifying the number of small nuclei positive for Histone-RFP and Prospero (Appendix Fig S2Q). Next, we specifically eliminated the expression of *Dronc* in EBs using the *Su(H)-Gal4* driver (Zeng *et al*, 2010). Unexpectedly, these experiments also failed to induce significant gut hyperplasia and penetrant differentiation phenotypes (lack of GFP in *Su(H)-expressing* cells and retention of Histone-RFP) in most the analysed intestines (Fig 3E–G). However, the EBs

were significantly increased in size (Fig 3H), possibly indicating the initiation of the EC differentiation programme. By comparison with the results obtained using the *esg*-Gal4 driver, these new data show that Dronc deficiency appears to sensitise the EBs to differentiate but mainly when its expression is compromised in both progenitor cells (see Discussion).

## Dronc functions rely on its preferential accumulation and activation in progenitor cells

To investigate the molecular origin of the Dronc phenotypes, we analysed the transcriptional activation of Dronc in the posterior midgut, using a newly created $Dronc^{KO-Gal4}$ strain (Fig EV2N). This fly line expresses Gal4 under the physiological regulation of the *Dronc* promoter. Interestingly, $Dronc^{KO-Gal4}$ expression was not

restricted to the progenitor cells (Appendix Fig S3A). We next investigated the *Dronc* protein levels in progenitor cells using a published *Dronc* allele endogenously tagged with the biotin-ligase TurboID (Shinoda *et al*, 2019). This enzyme reduces the threshold of detection of any protein of interest by transferring biotin to other proteins in close proximity (Shinoda *et al*, 2019). Interestingly, this construct revealed a restricted enrichment of biotinylation signal in small progenitor cells under our experimental conditions (Fig 4A). We validated this result taking advantage of a new reporter line created in the laboratory that expresses a catalytically inactive (C318A) tagged form of Dronc under the regulation of the *Actin* promoter (Appendix Fig S3B). This construct generates fertile adult flies without any noticeable developmental or morphological defects, and the tagging at the C terminus with a modified GFP and Myc facilitates its immunodetection (Appendix Fig S3C).

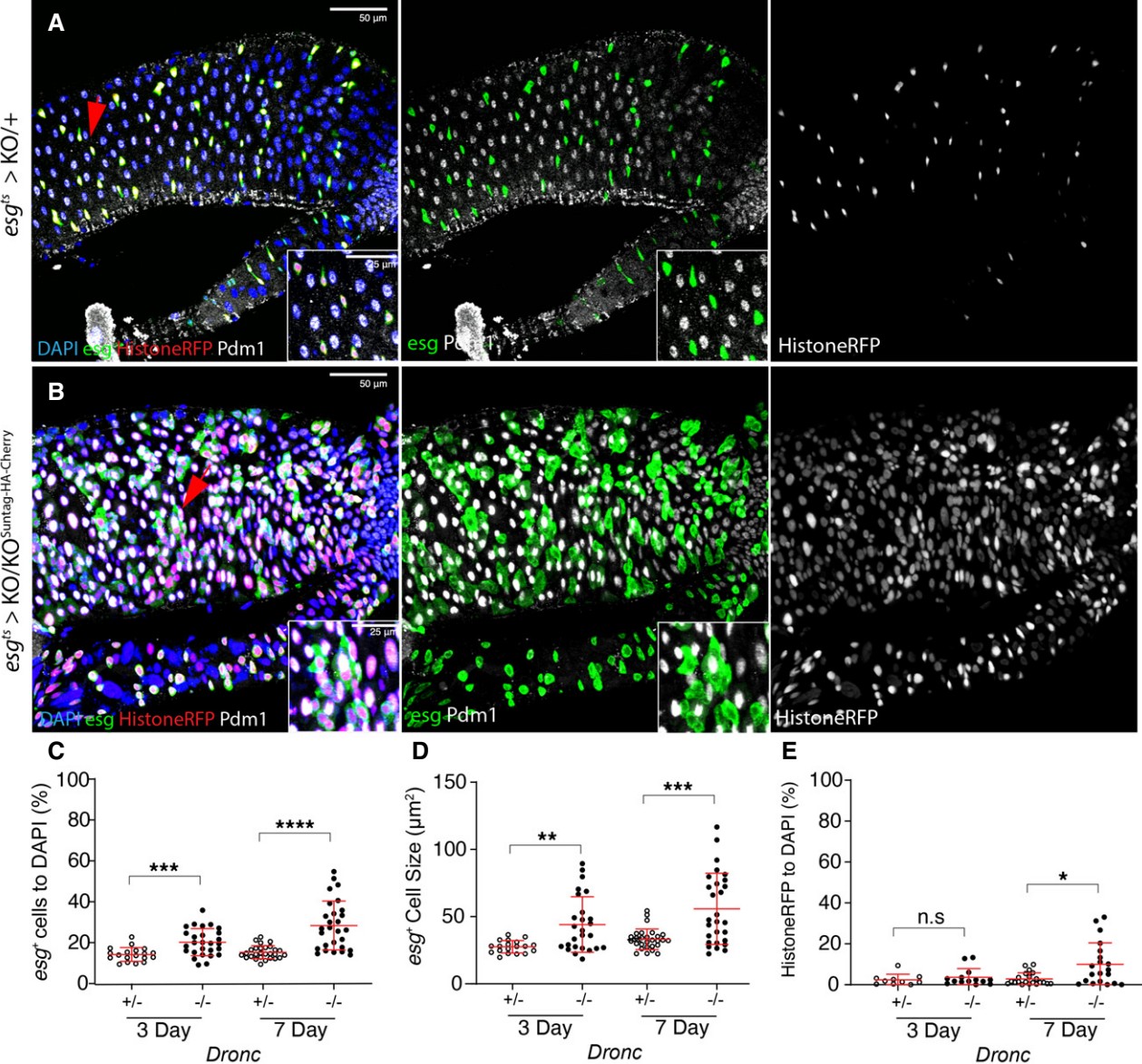

**Figure 2.**

◄

**Figure 2.  Effects of *Dronc* deficiency in intestinal stem cells in experimental conditions without basal cellular turnover.**

A   ReDDM activation in *Dronc* heterozygous (+/−) intestine 7 days after temperature shift at 29°C; *esg* expression (green) labels the intestinal progenitor cells, Histone-RFP (red) is a semi-permanent marker retained in differentiated cells and Pdm-1 (grey) labels differentiated ECs. The red arrows indicate the enlarged areas depicted in the insets in the entire figure. DAPI labels the DNA In the entire Figure (blue). All of the experiments in the entire figure have been made under experimental conditions that protect the epithelial integrity (intestine reared in Oxford Medium and flies transferred every 2 days into vials with fresh food). Genotype: $w^{1118}$; *esg*-Gal4 UAS-*CD8-GFP*/+; *TubG80*$^{ts}$ UAS-*Histone-RFP Dronc*$^{KO}$ (recombinant chromosome made for this study)/+.

B   ReDDM activation in a *Drosophila* intestine in which all of the progenitor cells are fully mutant for *Dronc* (see full genotype in Materials and Methods); note the increased cell size of *esg*-expressing cells (GFP+, green), the co-expression of Histone-RFP (red) and the EC marker Pdm-1 (grey), and the co-expression of Pdm-1 and GFP in enlarged cells (compare panels and detailed insets from A with B). The image is a representative example of a *Drosophila* adult gut 7 days after temperature shift at 29°C. Genotype: $w^{1118}$; *esg*-Gal4 UAS-*CD8-GFP*/+; *TubG80*$^{ts}$ UAS-*Histone-RFP Dronc*$^{KO}$/UAS-*Flippase* (BL8209) FRT *Dronc-GFP-APEX* FRT *suntag-HA-Cherry*.

C   Relative number of *esg*-expressing cells normalised to DAPI; notice that the relative percentage *of esg*-labelled cells is significantly higher in *Dronc* fully mutant conditions (−/−) at 3 days (***$P = 0.0007$) and 7d (****$P < 0.0001$) post-temperature shift at 29°C (Quantifications were made using $N \geq 2$ biological replicates; unpaired two-tailed *t*-test, +/− $n = 32$, −/− $n = 25$). Error bars represent standard deviation of the mean. Genotypes: +/−: $w^{1118}$; *esg*-Gal4 UAS-*CD8-GFP*/+; *TubG80*$^{ts}$ UAS-*Histone-RFP Dronc*$^{KO}$/+; −/−: $w^{1118}$; *esg*-Gal4 UAS-*CD8-GFP*/+; *TubG80*$^{ts}$ UAS-*Histone-RFP Dronc*$^{KO}$/UAS-*Flippase* (BL8209) FRT *Dronc-GFP-APEX* FRT *suntag-HA-Cherry*.

D   Average cell size of *esg*-expressing cells (μm²); note the increased cell size of *Dronc* fully mutant progenitor cells (−/−) at 3d (**$P = 0.0014$) and 7d (***$P = 0.0007$) post-temperature shift at 29°C (Quantifications were made using $N \geq 2$ biological replicates; unpaired two-tailed parametric *t*-test, 3d +/− $n = 19$, 3d −/− $n = 28$, 7d +/− $n = 29$, 7d −/− $n = 28$). Error bars represent standard deviation of the mean. Genotypes: +/−: $w^{1118}$; *esg*-Gal4 UAS-*CD8-GFP*/ +; *TubG80*$^{ts}$ UAS-*Histone-RFP Dronc*$^{KO}$/+; −/−: $w^{1118}$; *esg*-Gal4 UAS-*CD8-GFP*/ +; *TubG80*$^{ts}$ UAS-*Histone-RFP Dronc*$^{KO}$/ UAS-*Flippase* (BL8209) FRT *Dronc-GFP-APEX* FRT *suntag-HA-Cherry*.

E   Relative number of *esg*-negative cells expressing Histone-RFP normalised to DAPI; notice that the number of Histone-RFP cells without *esg* expression is significantly higher in *Dronc* fully mutant (−/−) conditions at 7d (*$P = 0.0046$) (Quantifications were made using $N$ biological replicates $\geq 2$; Mann–Whitney test, +/− $n = 24$, −/− $n = 20$. Error bars represent standard deviation of the mean. Genotypes: +/−: $w^{1118}$; *esg*-Gal4 UAS-*CD8-GFP*/+; *TubG80*$^{ts}$ UAS-*Histone-RFP Dronc*$^{KO}$/+; −/−: $w^{1118}$; *esg*-Gal4 UAS-*CD8-GFP*/+; *TubG80*$^{ts}$ UAS-*Histone-RFP Dronc*$^{KO}$/UAS-*Flippase* (BL8209) FRT *Dronc-GFP-APEX* FRT *suntag-HA-Cherry*.

Data information: (A, B) The red arrows indicate the enlarged areas depicted in the insets in the entire figure.

Interestingly, our construct was mainly accumulated in progenitor cells (Fig 4B–E) and preferentially enriched within EBs (*Su(H)*-LacZ-positive cells; compare Fig 4C with E). Expectedly, after inducing tissue damage, the restricted pattern was lost and the Dronc-GFP-myc was widely accumulated in all of the intestinal cell types (Appendix Fig S3D). Considering these findings, we combined the DBS-S-QF sensor with the EB marker *Su(H)*-LacZ. This experiment showed the prevalent activation of DBS-QF in EBs (Fig 4F and Appendix Fig S3E). Together, these findings correlated the functions of Dronc in progenitor cells with its specific accumulation and transient activation.

**Dronc activation limits Notch signalling**

Moderate-high levels of Notch signalling facilitate the entry of intestinal progenitor cells into several differentiation programmes (Hakim *et al*, 2010; Kapuria *et al*, 2012; Zhai *et al*, 2017). Since *Dronc* deficiency facilitates the conversion of EBs into ECs, we assessed the genetic interplay between Notch signalling and *Dronc*. To this end, we simultaneously eliminated the expression of *Dronc* and *Notch* in progenitor cells using the *esg*-Gal4 driver. This genetic manipulation induced phenotypes comparable to that of *Notch* deficiency alone, and therefore, the increase in cell size and EC features linked to *Dronc* deficiency were suppressed (compare Fig 5A–C with Figs 2B–E and EV3), even though we did notice a modest increase in the number of progenitor cells (Fig 5C). Conversely, the ectopic activation of the Notch pathway ($N^{intra}$) in *Dronc*-mutant progenitor cells caused the rapid elimination of intestinal precursors from the epithelia (Fig 5D and E). Furthermore, the reduction of intestinal precursor was more pronounced than when $N^{intra}$ was expressed alone (Fig 5F). In parallel to these genetic interactions, we observed that the transcriptional Notch-signalling reporter, *Su(H)*-LacZ, remained robustly upregulated within differentiated ECs mutant for *Dronc*, whilst paraquat treatment failed to replicate these results (compare Fig 5G–I). These data genetically located the function

of Dronc upstream of the Notch pathway, likely acting as a negative regulator.

# Discussion

### Non-apoptotic Dronc activation contributes to ensuring gut epithelial homeostasis

Our caspase lineage-tracing experiments (Baena-Lopez *et al*, 2018a) and genetic manipulations have suggested a non-apoptotic role for caspase activation in the intestinal progenitor cells, under non-regenerative conditions. Supporting this conclusion, the lack of essential apoptotic regulatory components (e.g. the terminal effector caspases), failed to alter the intestinal epithelial features under our experimental regime (Fig EV4). On the contrary, Dronc activation limits the number of progenitor cells, and their unintended conversion from EBs into ECs (Figs 2 and EV3). These findings support the hypothesis that Dronc contributes to the preservation of intestinal progenitor cells in quiescence state through mechanisms independent of apoptosis. Importantly, *caspase-9* deficiency in human intestinal precursors results in excessive proliferation and poor differentiation (Xu *et al*, 2013). Therefore, our findings could be relevant to understanding the origin of human intestinal malignancies. Beyond the implications in the intestinal system, our results confirm the non-apoptotic and caspase-dependent regulation of stem cell function (Aram *et al*, 2017; Bell & Megeney, 2017; Baena-Lopez *et al*, 2018b).

### Caspases can modulate the intestinal epithelial properties through non-apoptotic and apoptotic molecular mechanisms

Several reports have shown the ability of caspases to modulate the proliferation and differentiation of intestinal precursors in regenerative conditions (Amcheslavsky *et al*, 2009; Jin *et al*, 2013; Jin *et al*,

2017). In particular, the cleavage of the chromatin regulator Brahma mediated by the effector caspases restrains the proliferation and differentiation of ISCs after tissue damage (Jin *et al*, 2013). Here, we show that *Dronc* is required to prevent the increase in number of intestinal precursor cells as well as their unintended differentiation in non-regenerative conditions. These roles of *Dronc* are unlikely to be correlated with expression of Brahma, since the effector caspases are largely dispensable in our experimental setting (Fig EV4).

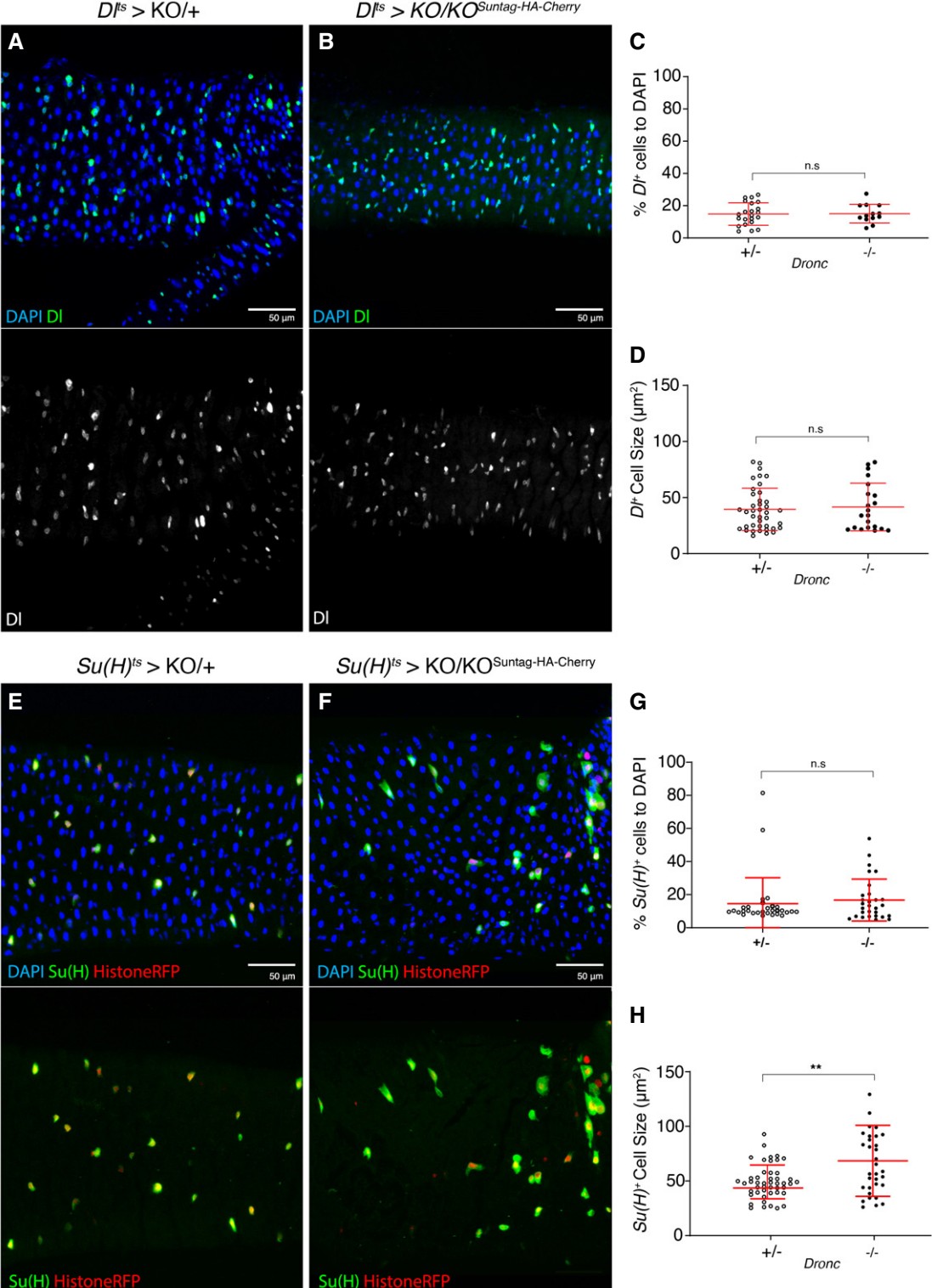

**Figure 3.**

**Figure 3. Phenotypic analysis of *Dronc* deficiency in *escargot*- and *Delta*-expressing cells.**

A  A 7d intestine *Dronc* heterozygous (+/−) intestine in which intestinal stem cells express GFP under the regulation of *Dl*-Gal4. All of the experiments described in the figure were performed in Oxford medium following an experimental regime that protects the epithelial integrity (flies transferred every 2 days into vials with fresh food). DAPI (blue) labels the DNA in the entire Figure. Genotype: $w^{1118}$;;*delta*-Gal4 UAS-*GFP TubG80$^{ts}$* UAS-*Histone-RFP Dronc$^{KO}$*/+ (recombinant chromosome made for this study; the parental line *delta*-Gal4 UAS-*GFP TubG80$^{ts}$* UAS-*Histone-RFP* was obtained from Maria Dominguez).

B  A 7d intestine in which the Intestinal Stem Cells have become *Dronc* homozygous mutant (−/−) upon expressing UAS-*Flp* under the regulation of *Dl*-Gal4. *Dl*-expressing cells are labelled with UAS-GFP (green); there are no morphological differences between (A) and (B). Genotype: $w^{1118}$;;*delta*-Gal4 UAS-*GFP TubG80$^{ts}$* UAS-*Histone-RFP Dronc$^{KO}$*/ UAS-*Flippase* (BL8209) FRT *Dronc-GFP-APEX* FRT *suntag-HA-Cherry*.

C  Relative number of GFP + *Delta*-expressing cells normalised to DAPI; there is no significant increase in intestinal stem cell number between heterozygous and homozygous *Dronc* mutant conditions (ns; $P = 0.9231$) (Quantifications were made using $N \geq 2$ biological replicates; unpaired two-tailed $t$-test, +/− $n = 22$, −/− $n = 13$). Error bars represent standard deviation of the mean. Genotypes: +/−:$w^{1118}$;;*delta*-Gal4 UAS-*GFP TubG80$^{ts}$* UAS-*Histone-RFP Dronc$^{KO}$*/ UAS-*Flippase* (BL8209) FRT *Dronc-GFP-APEX* FRT *suntag-HA-Cherry*; −/−: $w^{1118}$;;*delta*-Gal4 UAS-*GFP TubG80$^{ts}$* UAS-*Histone-RFP Dronc$^{KO}$*/UAS-*Flippase* (BL8209) FRT *Dronc-GFP-APEX* FRT *suntag-HA-Cherry*.

D  Average *Delta* cell size ($\mu m^2$); notice that the cell size does not change between heterozygous and homozygous *Dronc* mutant ISCs (ns; $P = 0.9694$) (Quantifications were made using $N \geq 2$ biological replicates; unpaired two-tailed $t$-test, +/− $n = 42$, −/− $n = 21$). Error bars represent standard deviation of the mean. Genotypes: +/−:$w^{1118}$;;*delta*-Gal4 UAS-*GFP TubG80$^{ts}$* UAS-*Histone-RFP Dronc$^{KO}$*/UAS-*Flippase* (BL8209) FRT *Dronc-GFP-APEX* FRT *suntag-HA-Cherry*; −/−: $w^{1118}$;;*delta*-Gal4 UAS-*GFP TubG80$^{ts}$* UAS-*Histone-RFP Dronc$^{KO}$*/UAS-*Flippase* (BL8209) FRT *Dronc-GFP-APEX* FRT *suntag-HA-Cherry*.

E  A 7d intestine *Dronc* heterozygous (+/−) intestine in which intestinal stem cells express GFP under the regulation of *Su(H)*-Gal4. Genotype: $w^{1118}$; *Su(H)*-Gal4 UAS-*GFP*; *TubG80$^{ts}$* UAS-*Histone-RFP Dronc$^{KO}$*/+.

F  A 7d intestine in which the Enteroblasts have become *Dronc* homozygous mutant (−/−) upon expressing UAS-*Flp* under the regulation of *Su(H)*-Gal4. *Su(H)*-Gal4-expressing cells are labelled with UAS-GFP (green) and UAS-Histone-RFP; note the increase in cell size when compared to (E) and the absence of Histone-RFP cells without GFP signal. Genotype: $w^{1118}$; *Su(H)*-Gal4 UAS-*GFP*; *TubG80$^{ts}$* UAS-*Histone-RFP Dronc$^{KO}$*/ UAS-*Flippase* (BL8209) FRT *Dronc-GFP-APEX* FRT *suntag-HA-Cherry*.

G  Relative number of *Su(H)*-Gal4-expressing cells normalised to DAPI; there is no significant increase in intestinal stem cell number between heterozygous and homozygous *Dronc* mutant conditions (n.s.; $P = 0.5099$) (Quantifications were made using $N \geq 2$ biological replicates; Mann–Whitney test, +/− $n = 30$, −/− $n = 29$). Error bars represent standard deviation of the mean. Genotypes: +/−: $w^{1118}$; *Su(H)*-Gal4 UAS-*GFP*; *TubG80$^{ts}$* UAS-*Histone-RFP Dronc$^{KO}$*/+; −/−: $w^{1118}$; *Su(H)*-Gal4 UAS-*GFP*; *TubG80$^{ts}$* UAS-*Histone-RFP Dronc$^{KO}$*/UAS-*Flippase* (BL8209) FRT *Dronc-GFP-APEX* FRT *suntag-HA-Cherry*.

H  Average *Su(H)* cell size ($\mu m^2$), there is a significant increase in *Su(H) Dronc* null cells (−/−) when compared to the heterozygous control condition (\*\*$P = 0.0064$) (Quantifications were made using $N \geq 2$ biological replicates; Mann–Whitney test, +/− $n = 49$, −/− $n = 33$). Error bars represent standard deviation of the mean. Genotypes: +/−: $w^{1118}$; *Su(H)*-Gal4 UAS-*GFP*; *TubG80$^{ts}$* UAS-*Histone-RFP Dronc$^{KO}$*/+; −/−: $w^{1118}$; *Su(H)*-Gal4 UAS-*GFP*; *TubG80$^{ts}$* UAS-*Histone-RFP Dronc$^{KO}$*/UAS-*Flippase* (BL8209) FRT *Dronc-GFP-APEX* FRT *suntag-HA-Cherry*.

Alternatively, they appear to be linked to the Dronc-dependent modulation of Notch signalling (Fig 5). This conclusion is supported by the upregulation of *Notch* target genes (e.g. *Su(H)*) in *Dronc*-mutant conditions and classical genetic interaction experiments (Fig 5). A physical interaction between Dronc and the Notch pathway inhibitor referred to as Numb was previously reported in the *Drosophila* nervous system (Ouyang *et al*, 2011); however, the genetic manipulation of this factor in our cellular scenario did not cause noticeable phenotypes (Appendix Fig S3F and G). This was expected to some extent since numb only participates in the conversion of ISCs into EEs (Salle *et al*, 2017). Future studies focused on investigating whether Dronc is able to alter Notch protein features (e.g. stability, internalisation…) or the expression of alternative Notch target genes (e.g. members of the *Enhancer of Split* family) could better define the interplay between this signalling pathway and Dronc. Beyond the novel non-apoptotic functions, caspases can also limit the number of progenitor cells generated in response to different stimuli via the apoptosis programme (Reiff *et al*, 2019), thus preventing gut hyperplasia in regenerative conditions. Together, these data illustrate the diversity of apoptotic and non-apoptotic caspase-dependent mechanisms ensuring intestinal homeostasis.

## Dronc activation helps to preserve the intestinal progenitor cells in quiescence

Our experiments have revealed that the elimination of *Dronc* expression in each type of intestinal progenitor cell do not cause major alterations in the intestinal epithelium (Fig 3H). However, the simultaneous elimination of *Dronc* in both progenitor cells (ISC

and EBs) robustly increases their number and induces the conversion of EBs into ECs (Fig 2). These data suggest that both types of progenitor cells require Dronc activation but having it in one of them suffices to maintain the intestinal epithelium in quiescence; therefore, Dronc could modulate the coordinated activity of progenitor cells. Our data also indicate that the functional requirement of Dronc is not equivalent in both progenitor cells and is highly correlated with its level of expression in each type of progenitor cell (Fig 4). From a molecular perspective, the coordinated behaviour of progenitor cells has been correlated with genetic factors expressed in both cell types [e.g. Sox protein family (Meng & Biteau, 2015; Zhai *et al*, 2015; Chen *et al*, 2016; Zhai *et al*, 2017; Meng *et al*, 2020)]. The upregulation of Sox proteins initiates signalling feedback loops that reactivate intestinal precursors, and EC differentiation in either homeostatic or stress conditions (Meng & Biteau, 2015; Zhai *et al*, 2015; Chen *et al*, 2016; Zhai *et al*, 2017). Furthermore, Sox expression can increase the activation levels of Notch signalling through the upregulation of Delta in ISCs, whilst facilitating the conversion of EBs into ECs (Zhai *et al*, 2017). Exploring the interplay between caspases and these factors could provide novel insights into the molecular implementation of caspase functions in the future. Beyond the Sox protein family, caspases could modulate cell–cell interactions between progenitor cells (Suzanne & Steller, 2009), the cytoskeleton dynamics and trafficking events (Duclos *et al*, 2017), thus influencing the activation levels of numerous signalling routes, including Notch (Zhai *et al*, 2017). Finally, *Dronc* could be buffering signalling fluctuations in different pathways (e.g. Notch) triggered by diverse intracellular, developmental or environmental stimuli (Khalil *et al*, 2014; Bell & Megeney, 2017; Weaver *et al*, 2020) (e.g. toxic agents, dietary conditions, infection). At this

point, the experimental validation of these and other hypotheses is beyond reach of this manuscript; however, we can conclude that non-apoptotic Dronc activation modulates Notch signalling and the coordination of intestinal progenitor cells. Importantly, although

Dronc appears to prevent the proliferation and differentiation of progenitor cells, its deficiency is not sufficient to reactivate these cells since not all of the *Dronc* mutant intestines displayed phenotypic manifestations. Therefore, we propose that caspases could be

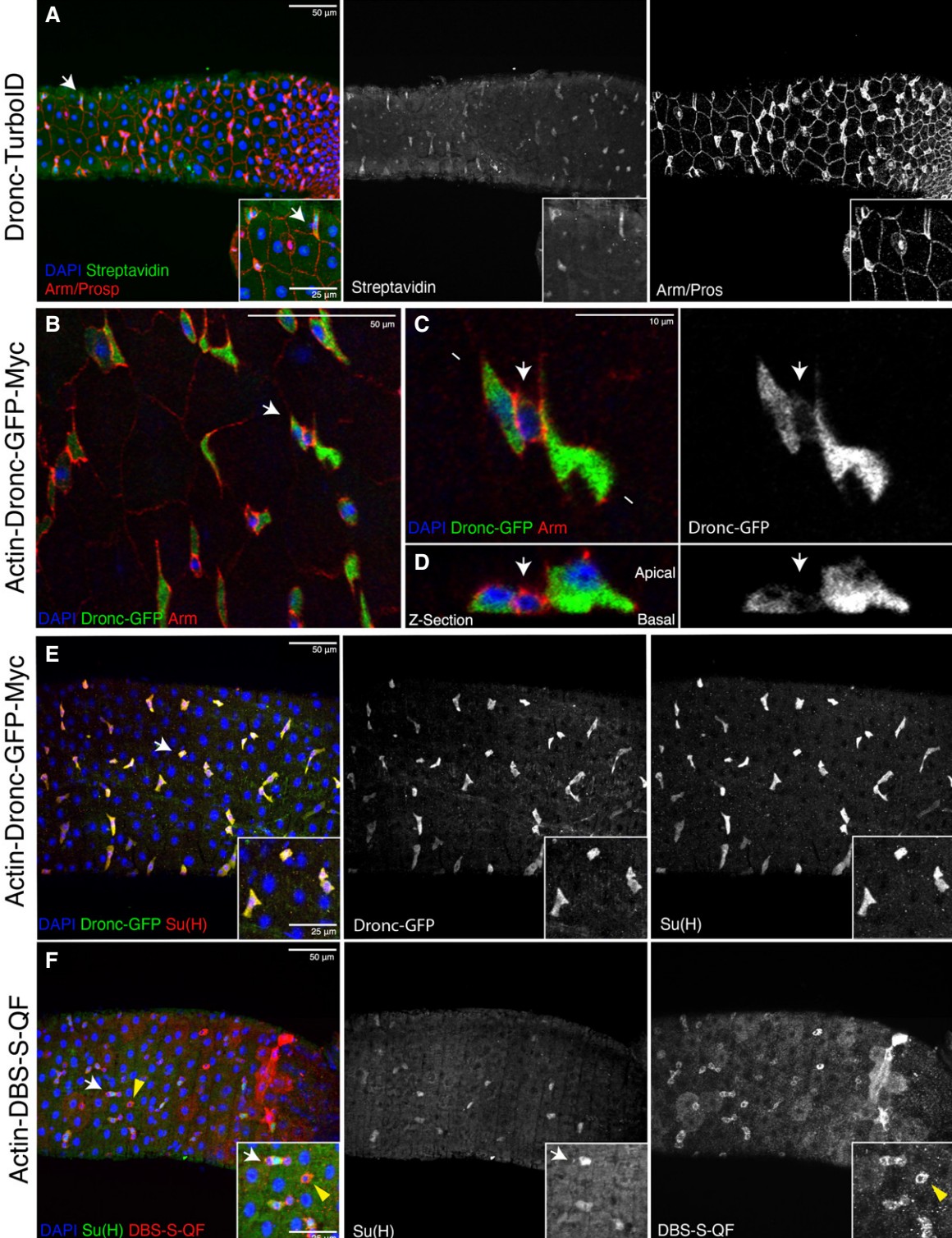

**Figure 4.**

◄

**Figure 4.   Dronc is preferentially accumulated and activated in progenitor cells.**

A   Representative image of a *Drosophila* posterior midgut showing the expression of Armadillo (red membranes), Prospero (red nuclei of EEs) and Dronc-TurboID (green shows Streptavidin-labelled proteins by Dronc-TurboID); note the preferential enrichment of Streptavidin staining is preferentially enriched in the intestinal progenitor cells (white arrows indicate an intestinal progenitor cell upregulating Dronc). Genotype: *Dronc-TurboID-V5 (Masayuki Miura)/Tm6B*.

B   Representative image of a *Drosophila* posterior midgut showing the expression of Armadillo (red membranes) and Actin-Dronc-GFP-Myc (green, anti-GFP); note the preferential enrichment of Dronc expression in the small groups of cells formed by intestinal progenitor cells (arrow). Genotype: $w^{1118}$; *Actin-Dronc*-GFP-Myc (attP-VK37)/Cyo.

C   High magnification of the picture shown in (B). The white arrow indicates a presumptive ISC showing low levels of Dronc compared with the flanking GFP + EBs. Genotype: $w^{1118}$; *Actin-Dronc*-GFP-Myc (attP-VK37)/Cyo. The arrow indicates a presumptive small ISC expressing high levels of Arm and low levels of Dronc-GFP-Myc.

D   Transversal Z-section of the cells shown in C. Genotype: $w^{1118}$; *Actin-Dronc*-GFP-Myc (attP-VK37)/Cyo. The arrow indicates a presumptive small ISC expressing high levels of Arm and low levels of Dronc-GFP-Myc.

E   The enteroblast marker Su(H) (red, immunostaining against Beta-galactosidase) strongly co-localises with high levels of Dronc expression (green, immunostaining against GFP); white arrow indicates the enlarged area depicted in the insets. Genotype: $w^{1118}$, *Su(H)GBE-LacZ* (Irene Miguel Aliaga); *Actin-Dronc*-GFP-Myc (attP-VK37)/Cyo.

F   There is extensive overlap between the expression of the EB marker Su(H) (green, immunostaining against Beta-galactosidase; white arrow) and the apical caspase reporter DBS-S-QF (red, immunostaining against HA), but it is also present in progenitor cells without Su(H) expression (yellow arrowheads). DAPI (Blue) labels cell nuclei in all the panels.

Data information: All of the experiments described in the figure apart from (F) were performed in Oxford medium following an experimental regime that protects the epithelial integrity at 29°C. The experiment shown in (F) was conducted at 25°C. Genotype: *Females $w^{1118}$Su(H)GBE-LacZ/y$^1$ $w^{1118}$ UAS-mCD8::GFP.L QUAS-mtdTomato-3xHA Act-DBS-S-QF*.

buffering the effect of other primary inducers promoting the exit from quiescence (e.g. environmental stress or intracellular signalling imbalance). Supporting this hypothesis, it has been shown that non-apoptotic caspase activation can ensure intracellular protein homeostasis (proteostasis) (Bell *et al*, 2016; Bell & Megeney, 2017) and can counteract the upregulation of signalling stress pathways (Fernando *et al*, 2002; Khalil *et al*, 2014; Bell *et al*, 2016; Weaver *et al*, 2020).

**Figure 5.   *Dronc* activation limits Notch signalling in the intestine.**                                                                        ►

A   Representative ReDDM labelling of a *Drosophila Dronc* heterozygous (+/−) intestine overexpressing an RNAi against Notch for 3 days; note the lack of fully differentiated Histone-RFP cells as EC (red) without *esg* expression (green, GFP). DAPI staining labels cell nuclei in panels A, B, D and E. All of the experiments described in the figure were performed in Oxford medium following an experimental regime that protects the epithelial integrity. Genotype: $w^{1118}$ UAS-*Notch-RNAi (Joaquin Navascués)*; esg-Gal4 UAS-CD8-GFP/+; TubG80$^{ts}$ UAS-*Histone-RFP Dronc*$^{KO}$/+.

B   Representative ReDDM labelling of a *Drosophila Dronc* mutant homozygous (−/−) intestine overexpressing an RNAi against Notch for 3 post-temperature shift at 29°C; note the lack of differentiated Histone-RFP cells as EC (red) without *esg* expression (green, GFP), as well as the increase of GFP-positive cells (compare A to B). Genotype: $w^{1118}$ UAS-*Notch-RNAi (Joaquin Navascués)*; esg-Gal4 UAS-CD8-GFP/+; TubG80$^{ts}$ UAS-*Histone-RFP Dronc*$^{KO}$/UAS-*Flippase* (BL8209) FRT *Dronc-GFP-APEX* FRT *suntag-HA-Cherry*.

C   Quantification of the number of Histone-RFP cells normalised to DAPI (proxy of progenitor cell proliferation obtained from the experiments shown in A and B panels); note the statically significant increase in Histone-RFP-positive cells in *Dronc* homozygous mutant intestines compared with controls (**$P$ = 0.0080) (Quantifications were made using $N \geq 2$ biological replicates; unpaired two-tailed *t*-test, +/− $n$ = 8, −/− $n$ = 15). Error bars represent standard deviation of the mean. Genotypes: +/−: $w^{1118}$ UAS-*Notch-RNAi*; esg-Gal4 UAS-CD8-GFP/+; TubG80$^{ts}$ UAS-*Histone-RFP Dronc*$^{KO}$/+; −/−: $w^{1118}$ UAS-*Notch-RNAi (Joaquin Navascués)*; esg-Gal4 UAS-CD8-GFP/+; TubG80$^{ts}$ UAS-*Histone-RFP Dronc*$^{KO}$/UAS-*Flippase* (BL8209) FRT *Dronc-GFP-APEX* FRT *suntag-HA-Cherry*.

D   *Drosophila Dronc* heterozygous intestine overexpressing the Notch intracellular domain for 7 days post-temperature shift at 29°C; intestinal *esg*-positive progenitor cells (green (GFP) and red (Histone-RFP). Genotype: $w^{1118}$; esg-Gal4 UAS-CD8-GFP/UAS-*Notch*$^{intra}$ (Joaquin Navascués); TubG80$^{ts}$ UAS-*Histone-RFP Dronc*$^{KO}$/+.

E   *Drosophila Dronc* homozygous intestine overexpressing the Notch intracellular domain for 7d post-temperature shift at 29°C; notice that the *Dronc* deficiency accelerates the elimination of intestinal progenitor cells induced by Notch overactivation (compare D and E). The white arrows indicate the position of insets 500 µm from the posterior region. Note the complete loss of *esg*-labelled cells in this region. Genotype: $w^{1118}$; esg-Gal4 UAS-CD8-GFP/ UAS-*Notch*$^{intra}$; TubG80$^{ts}$ UAS-*Histone-RFP Dronc*$^{KO}$/UAS-*Flippase* (BL8209) FRT *Dronc-GFP-APEX* FRT *suntag-HA-Cherry*.

F   Relative number of *esg*-positive cells to DAPI in either heterozygous or homozygous *Dronc*-mutant *esg* cells overexpressing Notch intra; note the significant reduction of *esg*-expressing cells (****$P$ < 0.0001) (Quantifications were made using $N \geq 2$ biological replicates; Mann–Whitney test, +/− $n$ = 17, −/− $n$ = 17). Error bars represent standard deviation of the mean. Genotypes: +/−: $w^{1118}$; esg-Gal4 UAS-CD8-GFP/UAS-*Notch*$^{intra}$; TubG80$^{ts}$ UAS-*Histone-RFP Dronc*$^{KO}$/+; −/−: $w^{1118}$; esg-Gal4 UAS-CD8-GFP/UAS-*Notch*$^{intra}$; TubG80$^{ts}$ UAS-*Histone-RFP Dronc*$^{KO}$/ UAS-*Flippase* (BL8209) FRT *Dronc-GFP-APEX* FRT *suntag-HA-Cherry*.

G   Representative image of a *Drosophila Dronc* heterozygous intestines 7 days after transfer to 29°C. *esg* expression in green (GFP) labels all intestinal progenitors cells, whilst Su(H)-lacZ (red) distinguishes a subpopulation of EBs within the *esg*-expressing cells. Genotype: $w^{1118}$, *Su(H)GBE-LacZ*; esg-Gal4 UAS-CD8-GFP/+; TubG80$^{ts}$ UAS-*Histone-RFP Dronc*$^{KO}$/+.

H   Representative image of a *Drosophila Dronc* KO intestine 7 days after transfer to 29°C. Note, the transcription of *Su(H)* in large cell GFP (−), suggestive of an aberrant upregulation of Notch signalling within fully differentiated ECs. Genotype: $w^{1118}$, *Su(H)GBE-LacZ*; esg-Gal4 UAS-CD8-GFP/+; TubG80$^{ts}$ UAS-*Histone-RFP Dronc*$^{KO}$/UAS-*Flippase* (BL8209) FRT *Dronc-GFP-APEX* FRT *suntag-HA-Cherry*.

I   Representative image of a 7d *Drosophila Dronc* heterozygous intestines following a 16-h treatment with paraquat. Note the absence of large *Su(H)* positive, GFP (−) cells compared with (H). The white arrows indicate the enlarged area depicted in the insets. Genotypes: +/−: $w^{1118}$, *Su(H)GBE-LacZ*; esg-Gal4 UAS-CD8-GFP/+; TubG80$^{ts}$ UAS-*Histone-RFP Dronc*$^{KO}$/+; −/−: $w^{1118}$, *Su(H)GBE-LacZ*; esg-Gal4 UAS-CD8-GFP/+; TubG80$^{ts}$ UAS-*Histone-RFP Dronc*$^{KO}$/UAS-*Flippase* (BL8209) FRT *Dronc-GFP-APEX* FRT *suntag-HA-Cherry*.

Data information: (D–H) The white arrows indicate the enlarged area depicted in the insets.

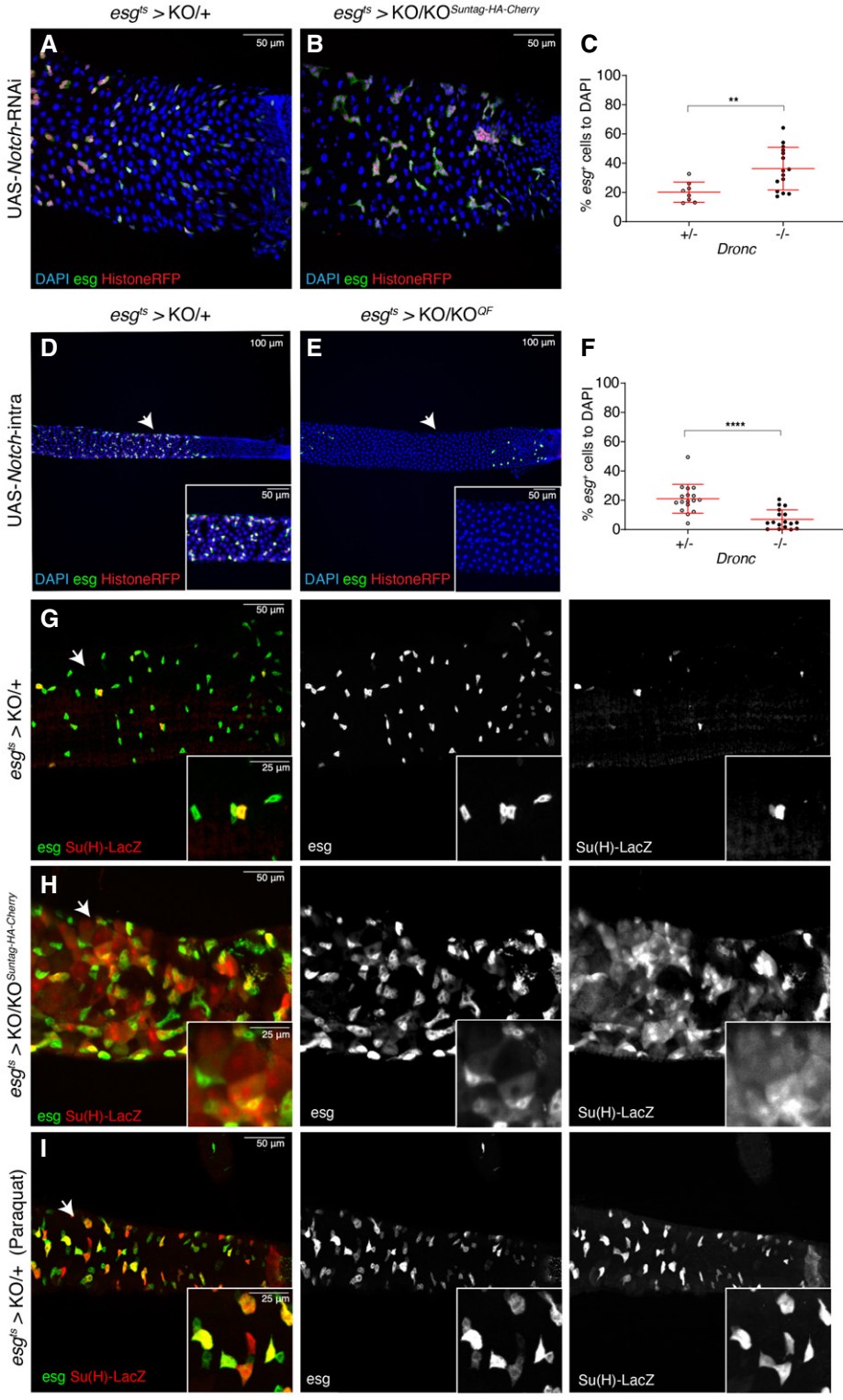

**Figure 5.**

## Tumour suppressor roles of caspases independent of apoptosis

Apoptosis is the main tumour suppressor mechanism linked to the caspases (Hanahan & Weinberg, 2011), and conceivably, sustained caspase activation can induce the elimination of intestinal progenitor cells (Reiff *et al*, 2019). However, caspases in the intestinal system display tumour suppressor activities independent of apoptosis. Along these lines, they can limit the proliferation and differentiation of progenitor cells (Jin *et al*, 2013; Lee *et al*, 2015) and our findings suggest that they help to preserve the progenitor cells

in quiescence. Collectively, these data support the hypothesis that caspases can act as tumour suppressor genes through non-apoptotic mechanisms.

# Materials and Methods

### Fly husbandry

All fly strains used are described at https://flybase.org/ unless otherwise indicated. Primary *Drosophila* strains and crosses were routinely maintained on Oxford fly food. The fly food has three basic components that should be properly mixed (food base, Nipagin mix and Acid mix). The Exact amount of the three components are described next.

Base composition per litre of fly food: agar (3 g/l, Fisher Scientific UK Ltd, BP2641-1), malt (64.3 g/l), molasses (18.8 g/l), maize (64.3 g/l), yeast (13 g/l), soya (7.8 g/l), water (1 l).

Nipagin mix per litre of fly food: Methyl-4-hydroxybenzoate (nipagin) (2.68 g, scientific labs, cat. W271004-10KG-K), absolute ethanol (25.1 ml, Fisher), water (1.3 ml).

Acid mix per litre of fly food: 5% phosphoric acid (Phosphoric Acid 85% Insect Cell Culture, cat. P5811-500G) in Propionic acid (5 ml, Propionic Acid Free Acid Insect Cell Culture, cat. P5561-1L).

Live yeast was added to each tube before transferring adult flies unless specified otherwise. Specific experiments also used Drosophila Quick Mix Medium Food (Blue) obtained from Blades Biological Ltd (DTS 070). 1 g of Drosophila Quick Mix Medium (Blue) was mixed with 3 ml of ddH$_2$O in these experiments.

### Conditional knockout and experimental regime during adulthood

Females were crossed with males at 18°C to prevent developmental lethality and transgene expression before adulthood. Adult female flies were then left mating with their siblings during 48 h and finally transferred to 29°C. At 29°C, the Gal80 repression of Gal4 is not effective allowing transgene expression. Until the dissection point, experimental specimens were transferred every 2 days to vials with fresh food containing yeast, or without yeast when using the Drosophila Quick Mix Media (Appendix Fig S1A).

### Paraquat treatment

The *Drosophila* were dry starved for 4 h. Subsequently, the flies were transferred to an empty fly vial in which the fly food was replaced by a flug soaked in a solution of 5% sucrose and 6.5 mM Paraquat. Flies were left in this vial for 16 h prior dissection.

### Molecular cloning and plasmid generation details

All PCRs were performed with Q5 High-Fidelity polymerase from New England Biolabs (NEB, M0492L). Standard subcloning protocols and HiFi DNA Assembly Cloning Kit (NEB, E5520S) were used to generate all the DNA plasmids. Transgenic lines expressing the new *Dronc* rescue constructs were obtained by attP/attB PhiC31-mediated integration. To this end, all the DNA plasmids were injected in *Drosophila* embryos containing the *Dronc*[KO]-reintegration site using Bestgene Inc. Fly strains generated will be

deposited at the Bloomington Stock Centre. Whilst resource transfer is completed, reagents will be provided upon request. Description of RIV-Gal4 and plasmid backbones used for generating the different DNA rescue constructs can be found in Baena-Lopez *et al* (2013).

### pTV-Cherry-Dronc[KO]. Dronc-targeting vector

We first amplified by PCR the homology arms for generating the gene targeting of *Dronc*. Genomic DNA extracted through standard protocols from $w^{1118}$ flies was used as a template. The sequences of the primers used are as follow:

5′ homology arm Forward primer with NotI restriction site: 5′ attatGCGGCCGCAAGTTGATGCAGCCTTCTGC 3′
5′ homology arm Reverse primer with KpnI restriction site: 5′ aattatGGTACCTCCGGTGACTCCGCTTATTGG 3′
3′ homology arm Forward primer with bglII restriction site: 5′ attgccaAGATCTACTGGACATTTTATCATTCC 3′
3′ homology arm Reverse primer with AvrII restriction site: 5′ attatggCCTAGGTCAAATCTGTTAATTTACG 3′

We then subcloned the PCR products into the targeting vector pTV-Cherry. The 5′ and 3′ homology arms were cloned into pTV-Cherry as a NotI-KpnI and BglII-AvrII fragments, respectively. The Dronc[KO] behaves as previously described null alleles of the gene and is homozygous lethal during pupal development. The molecular validation of the allele was also made by PCR. First, we extracted the DNA from 10 larvae using the Quick genomic DNA prep protocol described in the link below. http://francois.schweisguth.free.fr/protocols/Quick_Fly_Genomic_DNA_prep.pdf.

The sequence of the primers used for performing the PCR were:
Forward KO validation primer: 5′ TGAGCAGCTGTTGGCATTAGG 3′
Reverse KO validation primer: 5′ AGAGAATACCAATCATACTGC 3′

The PCR conditions were 30 s at 64°C of annealing temperature and 1 min at 72°C of extension. The PCRs were made using the Q5 High-Fidelity polymerase from New England Biolabs (NEB, M0492L).

### RIV-Dronc[KO-Gal4]

RIV-Gal4 plasmid was injected in the *Dronc*[KO]-reintegration site for generating the *Dronc*[KO-Gal4] line.

### RIV-Dronc[KO-Dronc WT-suntag-HA]

Two fragments were amplified by PCR using the full-length wild-type cDNA of *Dronc* as a template (LP09975 cDNA clone from Drosophila Genomics Resource Centre). This cDNA also contained the flanking untranslated regions of *Dronc* in 5′ and 3′. We also appended a suntag-HA-tag to the C-terminal end of the *Dronc* open reading frame, before the stop codon. The PCR fragments were then subcloned using HiFi DNA Assembly Cloning Kit into a PUC57 plasmid backbone previously opened with NotI-KpnI. The whole fragment was then transferred into the RIV [MCS-white (Baena-Lopez *et al*, 2013)] as a NotI-KpnI fragment. The plasmid was finally injected in the *Dronc*[KO]-reintegration site. The primer sequences used for generating the rescue plasmid are next indicated.

Forward primer 1: 5′ tctagAGGGGGCTACCCATACGACGTCCCTGA
CTATGCGTAAgctagcttgccgccactggacattttatcattccgg 3′

Forward primer 2: 5′ gacggccagtgcggccgcagatctcctaggccggaacgcgtgg
aagccatatccggaatgcagccgccggagctcgagattgg 3′

Reverse primer 1: 5′ tctagAGGGGGCTACCCATACGACGTCCCTG
ACTATGCGTAAgctagcttgccgccactggacattttatcattccgg 3′

Reverse primer 2: 5′ caaaaattagactcttttggccttagtcgggtaccggcgcgccactc
gagcactagtTCTGGCTGTGTatatactgg 3′

Importantly, flies expressing this construct are viable, fertile and morphologically normal. Only the adult wings are less transparent than normal and sometimes are not expanded normally.

### RIV-Dronc<sup>KO FRT-DroncWT-GFP-Apex-FRT QF</sup>. Conditional Dronc allele followed by QF

We first modified the WT cDNA of *Dronc* adding a GFP-Apex2 chimeric fragment. This fragment was appended in frame before the stop codon of *Dronc* in order to facilitate biochemical approaches not used in this manuscript. The cDNA of *Dronc* and the GFP-Apex2 chimeric fragments were amplified by PCR from the plasmids *dronc*<sup>KO-Dronc-WT</sup> and pCDNA-conexing-EGFP-APEX2 (addgene #49385), respectively. The sequences of the primers used to that end were:

Forward primer 1: 5′ tccggagcggccgcagatctCCGGAACGCGTGGAAGC
CATATCCG 3′

Forward primer 2: 5′ TCCCGGGTTTTTCAACGAAggcggaatggtgagc
aagggcgaggagc 3′

Reverse primer 1: 5′ tcgcccttgctcaccattccgccTTCGTTGAAAAACCC
GGGATTG 3′

Reverse primer 2: 5′ atgtccagtggcggcaagctagcttaggcatcagcaaacccaagc 3′

The PCR fragments were then subcloned using HiFi DNA Assembly Cloning Kit into RIV-*Dronc*<sup>KO-Dronc-WT-suntag-HA</sup> previously opened with BglII-NheI. The entire construct was finally transferred into RIV [FRT-MCS2-FRT QF; *pax*-Cherry (Baena-Lopez *et al*, 2013)] as a NotI-SpeI fragment. Notice that the extra sequences appended to *Dronc* (the FRT sites and the QF fragment) do not compromise the rescue ability of the construct. Indeed, these flies are identical to *RIV-Dronc*<sup>KO-Dronc-WT-suntag-HA</sup>. Homozygous flies expressing QF upon FRT-rescue cassette excision die during metamorphosis indicating this allele in such configuration behaves as previously described null alleles.

### RIV-Dronc<sup>KO FRT-DroncWT-GFP-Apex-FRT suntag-HA-Cherry</sup>. Conditional Dronc allele suntag-HA-Cherry

We subcloned a newly designed PCR product suntag-HA-Cherry in PUC57-*Dronc*<sup>KO-Dronc WT-suntag-HA</sup> by using HiFi DNA Assembly Cloning Kit. The PUC57-*Dronc*<sup>KO-Dronc WT-suntag-HA</sup> backbone vector was opened with AvrII-NsiI. The primers used for generating the suntag-HA-Cherry peptide were:

Forward primer 1: 5′gccagtgcggccGCagatctCCTAGGcccgggtttttcaacg
aaggggggcgaggagttgctgaGCAAAAATTATCATTTGGAGAacgaagtagcac
gactaaag 3′

Forward primer 2: 5′gtagcacgactaaagaaaggggtccggatcgggttctagagggggc
tacccatacgacgtccctgactatgcgGGGaattCCAACatggtgagcaagggcg 3′

Then, the suntag-HA-Cherry fragment was extracted from the new PUC57-*Dronc*<sup>KO-Dronc-WT-suntag-HA-Cherry</sup> vector a AvrII-blunted -- ClaI and transferred to *RIV-Dronc*<sup>KO FRT-DroncWT-GFP-Apex-FRT QF</sup> previously opened with AvrII-blunted -ClaI. Homozygous flies expressing suntag-HA-Cherry peptide under the physiological regulation of *Dronc* die during metamorphosis indicating this allele behaves as previously described null alleles.

### RIV-Dronc<sup>KO FRT-DroncWT-GFP-Apex-FRT Dronc-FLCA-suntag-HA-Cherry</sup> and RIV-Dronc<sup>KO FRT-DroncWT-GFP-Apex-FRT Dronc-FLCAEA-suntag-HA-Cherry</sup>. Conditional Dronc alleles FLCA and FLCAEA-suntag-HA-Cherry

We first generated two point mutations through gene synthesis (Genewiz) in the wild-type cDNA of *Dronc* that caused the following amino acid substitutions, C318A and E352A. This version of *Dronc* is enzymatically inactive (C318A) and cannot be either processed during the proteolytic activation steps of *Dronc* (E352A). This fragment was subcloned in PUC57-*Dronc*<sup>KO-Dronc-WT-suntag-HA-Cherry</sup> as a BglII-XmaI fragment, thus replacing the wild-type version of *Dronc* by the mutated. Finally, the DNA sequence was transferred to the *RIV-Dronc*<sup>KO FRT-DroncWT-GFP-Apex-FRT QF</sup> plasmid as an AvrII-ClaI fragment. Homozygous flies expressing this mutant form of *Dronc* die during metamorphosis indicating this allele behaves as previously described null alleles. An equivalent cloning strategy was followed to generate the FLCA allele, although in that case a single mutation (C318A) was introduced in the *Dronc* wild-type template through DNA synthesis.

### RIV-Dronc<sup>KO FRT-DroncWT-GFP-Apex-FRT Dronc-deltaCAEA-suntag-HA-Cherry</sup>. Conditional Dronc allele deltaCAEA-suntag-HA-Cherry

We generated a PCR product that deletes the CARD domain of Dronc using the following primers and as template for the PCR *RIV-Dronc*<sup>KO FRT-DroncWT-GFP-Apex-FRT Dronc-FLCAEA-suntag-HA-Cherry</sup>.

Forward primer 1: 5′ ggccagtgcggccGCCCTAGGGTTTaaac ggggaa
tgggcaattGtctggatgcggcc 3′

Reverse primer 1: 5′ catGTTGGaattccccgcatagtcagggacgtcgtatgggta
gcccccc 3′

The PCR product was subcloned in PUC57-*Dronc*<sup>KO-Dronc-suntag-HA-Cherry</sup> as a NotI-EcoRI fragment, thus inserting the truncated and catalytically inactive version of Dronc in frame with the Suntag-HA-Cherry peptide. Finally, the DNA sequence was transferred to the *RIV-Dronc*<sup>KO FRT-DroncWT-GFP-Apex-FRT QF</sup> plasmid as an AvrII-PasI fragment. Homozygous flies expressing this mutant form of *Dronc* die during metamorphosis indicating this allele behaves as previously described null alleles.

### RIV-Dronc<sup>KO FRT-DroncWT-GFP-Apex-FRT Dronc-FLWT-suntag-HA-Cherry</sup>. Conditional Dronc allele Dronc-FLWT-suntag-HA-Cherry

We generated a PCR product using as a template for the PCR *RIV-Dronc*<sup>KO-Dronc WT-suntag-HA</sup> and using the following primers.

Forward primer 1: 5′ ggccagtgcggccgcagatctcctaggccggaacgcgtggaagc
catatccggaatgcagccgccgga 3′

Reverse primer 1: 5′ cccttgctcaccatGTTGGaattCCCcgcatagtcagggac gtcgtatggg 3′

The PCR product was subcloned in PUC*57-Dronc*$^{KO\text{-}Dronc\text{-}suntag\text{-}HA\text{-}Cherry}$ as a NotI-EcoRI fragment, thus inserting the wild-type version of Dronc in frame with the suntag-HA-Cherry peptide. Finally, the DNA sequence was transferred to the *RIV-Dronc*$^{KO\ FRT\text{-}DroncWT\text{-}GFP\text{-}Apex\text{-}FRT\ QF}$ plasmid as an AvrII-PasI fragment. Heterozygous flies expressing this mutant form rescue the pupal lethality associated with *Dronc* insufficiency.

## Plasmid generation details of Actin-Dronc-CA-GFP-Myc

We first generated one point mutation through gene synthesis (Genewiz) in the wild-type cDNA of *Dronc* that causes the following amino acid substitution C318A. This version of *Dronc* is enzymatically inactive. Also, we appended to the C terminus a suntag and a HA peptide sequence in frame with the open reading frame (ORF) of Dronc. Downstream of the ORF was included 3′UTR of *Dronc* present in the genomic locus. Extra restriction sites were added at the 5′ and 3′ of the construct to facilitate future subcloning projects. The entire construct was subcloned in PUC*57* as a Not-KpnI fragment. We then opened this vector with SmaI and NheI; this enzymatic digestion eliminates the C-terminal tagging of Dronc (suntag-HA) whilst retaining the 3′UTR. Using HiFi DNA assembly, we inserted a PCR product that encodes for a modified version of GFP with a Myc tag appended at the C-terminal end. The primers used to amplify the modified GFP-Myc were:

Forward primer 1: 5′ GCTTTAATAAGAAACTCTACTTCAATcccggg tttttcaacgaagggggcATGATCAAGATCGCCACCAGGAAGTACC 3′
Forward primer 2: 5′ CGTGACCGCCGCCGGGATCACGGAAACCGA TGGCGAGCTGTTCACCGGGGTGG 3′
Reverse primer 1: 5′ CCACCCCGGTGAACAGCTCGCCATCGGTTTCC GTGATCCCGGCGGCGGTCACG 3′
Reverse primer 2: 5′ gataaaatgtccagtggcggcaagctagCttacaggtcctcctc gctgatcagcttctgctcGTTAGGCAGGTTGTCCACCCTCATCAGG 3′

The template used to obtain the GFP-Myc PCR product was extracted from genomic DNA of flies containing the construct UAS-GC3Ai (Schott *et al*, 2017). The construct was finally subcloned as a NotI-XhoI fragment in an Actin-polyA vector of the lab previously opened with NotI-PspXI. Sequence of the plasmid will be provided upon request until the vector is deposited in a public repository.

## Immunohistochemistry

Adult mated female *Drosophila* Intestines were dissected in ice-cold PBS. Following dissection, the intestines were immersed for 6 s in wash solution (0.7% NaCl, 0.05% Triton X-100) heated to approximately 90°C. Subsequently, the intestines were rapidly cooled in ice-cold wash solution. The intestines were then rapidly washed in PBT (0.3%) before blocking for at least 1 h in 1% BSA-PBT (0.3%). Primary antibodies were incubated overnight at 4°C and secondary antibodies at room temperature for 2 h, diluted in blocking solution. Primary antibodies used were Goat Anti-GFP (1:200, Abcam, ab6673) Chicken anti-Beta-galactosidase (1:200, Abcam, Ab9361); Rabbit Anti-HA (1:1,000, Cell Signalling, C29F4); Rabbit anti-Pdm1 (1:2,000, kind gift from Yu Cai), Mouse Anti-Armadillo (1:50, DSHB, N2 7A1

ARMADILLO-c); Mouse Anti-Prospero (1:20; DSHB, MR1A) and Rabbit Anti-P35 (1:100, Novus Biologicals, NB100-56153). The secondary antibodies used were DAPI (1:1,000, Thermo Scientific, 62248); Goat Alexa 488 anti-Chicken (1:200, Life technologies, A11039); Donkey Alexa-488 anti-Goat (1:200, Life Technologies, A1105); Donkey Alexa-555 anti-Rabbit (1:200, Life Technologies, A31572),Donkey Alexa-647 anti-Rabbit (1:200, Life Technologies, A31573); Donkey Alexa-555 Anti-Mouse (1:200, Life Technologies, A31570) and Donkey Alexa-647 Anti-chicken (1:200, Jackson, 703-605-155).

## EdU assay

Adult mated female *Drosophila* Intestines were dissected in ice-cold PBS. Following dissection, the intestines were transferred to an Eppendorf tube and incubated with 20 μM EDU (Click-iT™ EdU Cell Proliferation Kit for Imaging, Alexa Fluor™ 647 dye, C10340) in PBS for 60 min at room temperature. Following treatment, intestines were rapidly washed in PBS, before fixation in 4% paraformaldehyde (Thermo Fisher Scientific, 433689L) diluted in PBS for 1 h. The intestines were then washed before incubation with the Click-iT™ reaction cocktail according to the manufacturer's instructions. In order to avoid unspecific signal, the intestines were incubated with the blocking solution (1% BSA-PBT (0.3%)) at 4°C for at least 1 h. Primary antibodies were incubated at 2× concentration at 4°C overnight. Standard protocols were followed to complete the staining with secondary antibodies.

## Immunoprecipitation and Western blot

Twenty flies of each genotype were snap-frozen at −80°C, thawed, macerated on ice in 200 μl of RIPA buffer (complemented with protease inhibitor), cleared at 20k *g* for 20 min and supernatant toped-up with PbS + 0.3% Triton X-100 to 500 μl. 50 μl of each sample was set aside, and the remaining 450 μl incubated with 15 μl of HA magnetic beads (Pierce) pre-washed three times in PbS + 0.3% Triton X-100 head-over-head at 4°C on a spinning wheel for 90 min. Beads were washed three times in PbS + 0.3% Triton X-100. Bound material was eluded from the beads in 50 μl Laemmli buffer and the equivalent of the lysate of five flies loaded on a 4–12% gradient gel. Western blot was revealed with rabbit HA antibody (CST 1:1,000) or beta-actin (DSHB 1:500).

## Generation of genetic mosaics

Females *yw hs-Flp1.22; Sp/Cyo; FRT80A UbiGFP/TM6B* were either crossed with males *w*$^{1118}$; *Dronc I29 FRT80/TM6b* (Experimental) or *w*$^{1118}$; *+/+; FRT80 ry*$^+$/*FRT80 ry*$^+$ (Control). Flies were selected every 2 days and then heat shocked for 90 min at 37°C. Following the heat shock treatment, flies were returned to 25°C for 7 days until dissection.

## Imaging of fixed samples

Fluorescent imaging of the R5 posterior region (Dutta *et al*, 2015) of *Drosophila* intestines were performed using the Olympus Fluoview FV1200 and associated software. Z-stacks were taken using either the 40× or 10× lenses. Acquired images were processed and quantified using automated Fiji/ImageJ (Schindelin *et al*, 2012; Rueden *et al*, 2017) macros or manually when automatisation was not possible. Generally, Z-stacks were projected, and the channels split. The "despeckle" tool was utilised to remove noise. The image was then

"thesholded" and the "watershed" tool used to segment joined cells. To count the number of objects, the "Analyse Particles…" function was utilised in automated quantification, and for manual counting, "cell counter". Figures were produced with Adobe Illustrator CC 2017.

### Statistical analysis

Microsoft Excel was used to complete the basic numerical preparation of the data. The data were subsequently collated in GraphPad Prism (8). The "identity outlier" analysis was utilised using the ROUT method with a Q value of 1% (All $N$ numbers listed in figures are prior to this analysis). The cleaned data were then tested for normality using the D'Agostino-Pearson omnibus normality test except for qPCR data which were tested using the Shapiro–Wilk normality test. All subsequent analysis is referenced in Figure Legends. The $P$ value format used is as follows: ns = $P >$ 0.05, $*P \leq 0.05$, $**P \leq 0.01$, $***P \leq 0.001$ and $****P \leq 0.0001$.

### RNA extraction and cDNA synthesis of *Drosophila* intestines and qPCR

Around 20 adult mated female *Drosophila* intestines were dissected in ice-cold PBS. Following dissection, the intestines were transferred to autoclaved Eppendorf tubes containing 350 µl of RLT Buffer plus from the RNeasy Plus Micro Kit (QIAGEN Cat. No 74034) with 1% $v/v$ of 2-mercaptoethanol (Sigma-Aldrich M6250). The intestines were homogenised using a new 1.5 ml pestle (Kimble 749521-1590) for each sample. The RNA was then extracted using the protocol and materials outlined in the RNeasy Plus Micro Kit. RNA quantity in samples was assessed twice using the Nanodrop Lite Spectrophotometer (Thermo Fisher Scientific) and the average determined. 500 µg of RNA was synthesised to cDNA (Thermo Fisher Scientific Maxima First-Strand cDNA Synthesis Kit—K1671). QPCR was performed using reagents and protocols from QIAGEN QuantiTect SYBR® Green PCR Kit (Cat No./ID: 204145) and using the QIAGEN Rotor-Gene Q.

The primer sequences were:

Rpl32 (Gomez-Lamarca *et al*, 2018):
Forward: 5′ ATGCTAAGCTGTCGCACAAATG 3′
Reverse: 5′ GTTCGATCCGTAACCGATGT 3′
Alpha-Trypsin* (PD44223):
Forward: 5′ ATGGTCAACGACATCGCTGT 3′
Reverse: 5′ CTGGCTCTGGCTAACGATGT 3′
Amylase-D* (PD40005):
Forward: 5′ GCATAGTGTGCCTCTCCCTC 3′
Reverse: 5′ TACGACCGGATGCGTAGTTG 3′
Jon65Aiii (Bozler *et al*, 2017):
Forward: 5′ AACACCTGGGTTCTCACTGC 3′
Reverse: 5′ TCAGGGAAATGTCGTTCCTC 3′

*Primers were sourced from the QPCR FlyPrimerBank tool (Hu *et al*, 2013).

## Data availability

All the raw images and data sets use for quantification purposes in the manuscript will be distributed upon reasonable request. The new fly strains generated for this manuscript will be deposited in the Bloomington Drosophila Stock Center (https://bdsc.indiana.edu/) and distributed upon request whilst the deposit is processed. This study includes no data deposited in external repositories.

**Expanded View** for this article is available online.

### Acknowledgements
The authors would like to thank Irene Miguel-Aliaga, María Dominguez, Andreas Bergmann, Joaquín Navascués and Iswar Hariharan for sharing various flies and reagents; Yu CAI (Temasek Life Sciences Laboratory Limited) for generously sharing the Pdm1 antibody; Joaquín Navascués for generously sharing reagents and protocols; Antonello Zeus for his intellectual input at the initial stages of the project; and the Caspase Lab members for their critical reading of the manuscript and invaluable suggestions. This work has been supported by Cancer Research UK 560C49979/A17516 and the John Fell Fund from the University of Oxford 162/001. L.A.Baena-Lopez is a CRUK Career Development Fellow (C49979/A17516) and an Oriel College Hayward Fellow. J. Alonso was a CRUK postdoctoral scientist associated with the grant code previously described. F. Wendler is a CRUK postdoctoral scientist associated with the grant code previously described. D. Antoni. Nahotko was a summer student supported by the ERASMUS (+) programme. L. Arthurton was a PhD student supported by the Edward Penley Abraham Research Fund.

### Author contributions
Conceptualisation: LAB-L; Plasmid generation, molecular biology protocols, allele preparation: DAN, JA, LAB-L; Experimental design, discussion: LA, LAB-L; Experimental work: LA; Immunoprecipitation, Western blot: FW; Writing manuscript, correction: LA, LAB-L. Figures: LA, LAB-L; Comments, approval: all co-authors.

### Conflict of interest
All authors declare that they have no conflicts of interest.

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
