## [Review Process File · EMBO Reports]

Non-apoptotic caspase activation preserves *Drosophila* intestinal progenitor cells in quiescence

Lewis Arthurton, Dominik Antoni Nahotko, Jana Alonso, Franz Wendler and Luis Alberto Baena Lopez

DOI: 10.15252/embr.201948892

Corresponding author(s): Luis Alberto Baena Lopez (alberto.baenalopez@path.ox.ac.uk) , Luis Alberto Baena Lopez (alberto.baenalopez@path.ox.ac.uk)

Review Timeline:

Submission Date:	17th Jul 19
Editorial Decision:	14th Aug 19
Revision Received:	12th Feb 20
Editorial Decision:	9th Apr 20
Revision Received:	31st Jul 20
Editorial Decision:	21st Sep 20
Revision Received:	21st Sep 20
Accepted:	1st Oct 20

Transaction Report:

Dear Dr. Baena Lopez,

Thank you for the submission of your research manuscript to EMBO reports. We have now received the full set of referee reports that are copied below.

I am sorry to say that the decision on your manuscript is not a positive one. As you will see, none of the referees provides strong support for the publication of your work in EMBO reports. The referees indicate that the relevance of the findings is questionable and that the data is rather preliminary, and doubt the central conclusions. Further, they note technical shortcomings. As the reports are below, I will not detail them here.

Given the comments of the three referees, the amount of work required to address them, and the fact that EMBO reports can only invite revision of papers that receive enthusiastic support from the referees upon initial assessment, we cannot offer to publish your manuscript.

I am sorry to have to disappoint you this time. I nevertheless hope, that the referee comments will be helpful in your continued work in this area, and I thank you once more for your interest in our journal.

Yours sincerely,

Achim Breiling
Editor
EMBO Reports

Referee #1:

In this manuscript, the authors define a function of the initiator caspase Dronc in the *Drosophila* midgut. First, they demonstrate initiator caspase activity in progenitor cells which they later identify as enteroblasts (EBs). This caspase activity is independent of effector caspases and non-apoptotic in nature. They also show that under the experimental and feeding conditions, there is no cell turnover and no differentiation. To identify a non-apoptotic function of Dronc in the midgut, the authors generated a new mutant allele of Dronc (DroncKO) in which the first exon is replaced by an attP site. The attP site was used by the authors to insert various wild-type, mutant and tagged Dronc constructs for characterization. In this manner, they created a conditional Dronc allele that can be flipped out in every cell type depending on the tissue-specific Gal4 line. When they conditionally knocked-out Dronc in progenitor cells, they observed premature differentiation of EBs into enteroblasts (ECs) and hyperplasia. Using a tagged Dronc transgene, they showed that Dronc is expressed in EBs. Finally, they show that Dronc acts upstream of Notch and IR signaling in a negative manner.

This is a potentially interesting manuscript, but there a number of issues which need to be addressed by the authors before publication.

1. I am not sure why the authors examine the non-apoptotic function of Dronc in the R4/R5 region of the midgut. Looking at the figures, it appears more to be R5 than R4. That is problematic as R5 borders the hindgut which influences the behavior of R5 quite a bit. Why did the authors not

address this question in the R4ab compartment which is commonly used for midgut studies? Therefore, I am not so sure how relevant the findings in this manuscript are.

2. The authors rule out effector caspases for this function of Dronc. However, Tang et al. (2015) and Ding et al (2016) showed that there is also a lot of non-apoptotic effector caspase activity in the midgut which appears to be particularly high in the R5 region. Ding et al showed that this effector caspase activity is dependent on Dronc. How does this all work together? Can both systems be combined to determine if they are overlapping or independent?

3. The authors do not present a phenotypic characterization of the DroncKO allele, other than to say that it is pupal lethal with other Dronc alleles. Is this allele actually a loss-of-function allele? The reason why I am asking this question is that the authors only remove the first exon which encodes the prodomain of Dronc. The large and small catalytic subunits are encoded by the second exon which is still intact. There have been cases reported that when a small deletion removes the initiator ATG, an internal in-frame ATG can be used for translation initiation which can generate a gain-of-function mutant. Meyer et al. (2000) has shown that a prodomain-less Dronc construct has increased caspase activity. Can the authors exclude this possibility for DroncKO?

4. In Figure 2B, the authors observe hyperplasia. Can they confirm this with mitotic markers?

5. Figure 3. The authors generate a GFP/Myc-tagged Dronc construct under Actin promoter control. Given that this construct is driven under the ubiquitous Actin promoter, why is it only detectable in EBs? The fact that an exogenously expressed protein is expressed in EBs does not mean that the endogenous gene is expressed there, too. In fact, looking at the Dronc-Gal4 expression in sup Fig 5A, it appears to this author that the expression is largely in ECs.

6. Figure 4. The authors perform epistasis experiments with Dronc and Notch. It would be nice if they could confirm these assays with a Notch marker.

7. Regarding the control of Notch by Dronc, is there any genetic interaction between Dronc and Numb in the midgut. Do Numb mutants have a phenotype in the midgut?

8. In sup fig 3J and 3K, the authors express a C318A E352A mutant of Dronc. Due to the less penetrant phenotypes, the authors conclude that Dronc requires catalytic activity (blocked by C318A) for the non-apoptotic role in the midgut. However, this conclusion is not exact. E352A blocks the processing of Dronc into large and small subunits which might be the reason for the less penetrant phenotypes. Not sure why the authors did this experiment in an E352A background. It is also not clear why in sup fig 3K, they deleted the CARD domain.

Referee #2:

In this manuscript, the authors describe a non-apoptotic function of the initiator caspase Dronc in limiting the differentiation of enteroblasts under normal homeostasis. By using a number of novel genetic reagents, they establish the requirement of Dronc catalytic activity in enteroblasts, and not intestinal stem cells, to limit their differentiation into enterocytes. These findings are interesting as they further support the non-apoptotic functions of caspases and describes a cell-type specific function in the adult posterior gut in *Drosophila*

The authors suggest that the catalytic activity of Dronc is required rather than Dronc acting as a scaffold for protein interactions. This raises several questions including how is Dronc activated and how is apoptosis prevented in the presence of Dronc activity? While Dronc can undergo autoactivation, the requirement of Dark for Dronc activation should be assessed. On the basis of the DBS-S-QF sensor, the abundance and activation of Drice should also be assessed. Are there conditions where enteroblast differentiation can be blocked by Dronc overexpression or does this promote apoptosis? These issues need to be addressed and discussed in further detail.

In the current format, the findings are preliminary and require a more extensive evaluation that in the most part uses the reagents and methods they already have established. This should include a thorough analysis with appropriate controls of all of the cell types (e.g. esg, delta, Su(H), Pros and Pdm) as well as markers of proliferation and cell death, for each genotype examined. The quantitation analysis should be kept consistent across the experiments. In addition, the rationale for several experiments and the presentation order of the figures as well as the inclusion of supplemental data in the main figures could all be improved.

Referee #3:

In this paper, Arthurton et al. present data which they believe shows a function for the Caspase, Dronc, in enteroblast (EB) differentiation in the fly intestine. A non-apoptotic, differentiation function for a caspase in gut homeostasis is potentially quite interesting. The fly strains generated here, which have conditional alleles of Dronc, are elegant tools that should be generally useful for studying caspase function in diverse contexts. However, this paper has numerous technical flaws and shortcomings that led us to doubt its central conclusions. In addition there are many things that are not clearly presented. Specific comments follow below:

1. The introduction (lines 80-90) does not give an accurate or comprehensive summary of what the published literature reports about the mechanisms of the ISC-EB-EC differentiation pathway. For instance, well known roles for Esg, Pdm1, Stat, and su(H) are not mentioned. Since this paper is about EB differentiation, the introduction to this topic should be accurate.
2. Using the REDDM cell lineage tracing system, the authors make the interesting and important point that under their conditions, ISCs in the midgut can be essentially quiescent. However there are many papers that show how rates of ISC division change as a function of gut region, fly age, sex, and mating. The authors need to specify clearly what gut region, sex, and age of flies they've used for their experiments.
3. In figures 1A and 1D, it looks like that the authors only show the R5 region of the midgut. Based on these pictures, it is hard to tell that DBS-S-QF is strikingly upregulated upon paraquat treatment. Much of the upregulation most likely happened in the malpighian tubules. The authors need to show whole midgut pictures for figures 1A and 1D. It could also help to quantify the DBS-S-QF fluorescence values.
4. QF, which is activated in one of their conditional K/O strains, is known to be toxic for Drosophila cells. Thus the experiments with the QF-tagged Dronc allele may be misleading. The authors should either test this, or use the Suntag allele for all experiments (it is already used for most experiments).
5. Using the P35 overexpression data (Fig. 1G-1I) to suggest a non-apoptotic function of caspase in

the gut is quite indirect and not so informative. It might be better to test the rate of gut regeneration in homozygous Df(3L)H99 (reaper/hid/grim triple mutants) flies. Further, knockdown of Dronc in EBs in the H99^{-/-} background (apoptosis-deficient condition) should be more direct way to determine the non-apoptotic function of Dronc during EB-to-EC differentiation. The EB specific driver, su(H)-Gal4-TS can be used here. It is surprising this tool is not used in this manuscript.

6. In Fig 1, the authors should plot the frequencies of cells that are Red⁺ Green⁺, Red⁺ Green⁻, and Red⁻ Green⁺. This would be more informative.

7. In Figure 2, the authors should use the Su(H)GBE-Gal4, a well-defined EB-specific driver, to knock out Dronc. The results might also be validated using standard FRT or MARCM clones mutant for Dronc, using another allele. In addition, the figure is not clear. What is the full genotype of the samples shown in 2A, 2B? Where is Suntag-Cherry? Counts of mitoses could also be useful here (to confirm Fig 2C).

8. The data in Fig 2 consistent with hyperplasia and proliferation could be explained be a requirement for Dronc in cell viability, rather than differentiation. The authors have not done enough to rule out this possibility.

9. The expression analysis of Dronc is poor. Endogenous Dronc or tagged active Dronc could be detected for some un-explained reason. An inactive tagged for was detected (Fig 3), but it is not clear that this mirrors the actual expression of endogenous Dronc. Moreover, the double staining in Fig 3 suggests the Dronc may be expressed in ISCs as well as EBs. The legend to this figure is not clear about what is shown.

10. The epistasis tests shown in Figure 4 are not meaningful. These merely suggest that Notch signaling and mTOR are dominant to Dronc in regulating differentiation. These results don't really imply that Dronc acts "as a negative regulator of the terminal differentiation program", as the authors suggest. The authors at least need some biochemistry or molecular data to support that Dronc acts through Notch or mTOR pathway to regulate EB-to-EC differentiation. For instance, examining the expression levels of the critical components of each pathway upon Dronc LOF or GOF could support their thesis. As noted above (point #8), many of the results presented here are also consistent with the possibility that loss of Dronc promotes gut epithelial turnover because Dronc is somehow essential for cell viability.

** As a service to authors, EMBO Press provides authors with the ability to transfer a manuscript that one journal cannot offer to publish to another journal, without the author having to upload the manuscript data again. To transfer your manuscript to another EMBO Press journal using this service, please click on
Link Not Available

Referee: 1

1. I am not sure why the authors examine the non-apoptotic function of *Dronc* in the R4/R5 region of the midgut. Looking at the figures, it appears more to be R5 than R4. That is problematic as R5 borders the hindgut which influences the behavior of R5 quite a bit. Why did the authors not address this question in the R4ab compartment which is commonly used for midgut studies? Therefore, I am not so sure how relevant the findings in this manuscript are.

We focused our initial analysis in the final region of the R4 and R5 because we previously described a wave of non-apoptotic caspase activation upon adult eclosion starting from these regions, that later spreads into more anterior parts of posterior midgut (1). However, we have now expanded the study into the more anterior regions of the posterior midgut mentioned by the reviewer (R4ab). The new version of the manuscript now includes low magnification images of such regions. In Supplementary Fig 3, we include low magnification images of the posterior midgut in *Dronc* heterozygous and *Dronc* KO conditions. This data complements the low magnification images of the *Dronc* LOF and Notch intra experiments of Figure 4, illustrating the loss of *esg* expressing cells throughout the whole of the posterior midgut. Finally, we provide data showing that paraquat treatment induces regeneration in the whole posterior midgut, whilst there is no evidence of tissue replenishment in our experimental conditions. Collectively, this information confirms the suitability of our analysis in the R4/5 region and validates our conclusions in the entire posterior midgut. Based on this, we have to respectfully disagree with the referee regarding the relevance and validity of our findings.

2. The authors rule out effector caspases for this function of *Dronc*. However, Tang et al. (2015) and Ding et al (2016) showed that there is also a lot of non-apoptotic effector caspase activity in the midgut which appears to be particularly high in the R5 region. Ding et al showed that this effector caspase activity is dependent on *Dronc*. How does this all work together? Can both systems be combined to determine if they are overlapping or independent?

We would like to thank the reviewer for this interesting question. We have combined the Ding et al. reporter with our previously published apical sensor, DBS-S-QF (Supplementary Figure 1) (1). The DBS-S-QF reporter appears to be mostly activated within the small progenitor cells, whereas the Ding et al. reporter is mainly activated within ECs. Interestingly, while there are some instances of overlap between the reporters, it also appears that they can be activated independently. These differences could emerge from the fact that the activation of DBS-S-QF precedes the activation of CasExpress, since the first reports on initiator and subsequently effector caspase activation, whilst CasExpress is only activated by effector caspases. Additionally, this disparity between the reporters could result from either an alternative non-apoptotic functional requirement for different caspase members, or specific response thresholds of activation for each sensor. Nevertheless altogether this confirms the presence of non-

apoptotic caspase activation within the intestine, but untangling the function in each cell type is outside of the scope of this manuscript,

3. The authors do not present a phenotypic characterization of the *Dronc*KO allele, other than to say that it is pupal lethal with other *Dronc* alleles. Is this allele actually a loss-of-function allele? The reason why I am asking this question is that the authors only remove the first exon which encodes the prodomain of *Dronc*. The large and small catalytic subunits are encoded by the second exon which is still intact. There have been cases reported that when a small deletion removes the initiator ATG, an internal in-frame ATG can be used for translation initiation which can generate a gain-of-function mutant. Meyer et al. (2000) has shown that a prodomain-less *Dronc* construct has increased caspase activity. Can the authors exclude this possibility for *Dronc*KO?

We would like to thank the reviewer for their query and have sought to clarify this within the text. To answer the reviewer's questions, the *Dronc*KO allele contains a genomic deletion (see PCR analysis of the genomic region shown in Supplementary Figure 2B) which creates a null allele. This allele has the first exon removed and replaced with an attP landing site. As the reviewer can see in Supplementary Figure 2C, we obtain identical phenotypes using heteroallelic combinations of *Dronc* null alleles (e.g. *Dronc*^{I29}) with *Dronc*KO, and homozygous *Dronc*KO as previously described with other null alleles (pupal lethality, (2)). Furthermore, we were able to demonstrate that the phenotypes associated with our *Dronc*KO mutant can only be rescued by re-inserting wildtype *Dronc* cDNA into the *Dronc*KO landing site (Supplementary Figure 3). The functional rescue obtained with the wildtype cDNA of *Dronc* in our KO line prevents *Dronc* LOF lethality and fulfils the requirement of *Dronc* in specific tissues such as the gut). Whilst we understand the reviewers concern that our allele may act as a gain of function allele based on the findings of Meyer et al (2000), the data described discards that possibility and confirm the nature of our KO allele. Although this information was included in our original submission, in the new version of the text we have modified these points in order to avoid any confusion regarding the nature of our KO allele.

4. In Figure 2B, the authors observe hyperplasia. Can they confirm this with mitotic markers?

We have performed an EDU staining in *Dronc* heterozygous vs null backgrounds (see Supplementary Figure 3) this shows an increase in cell proliferation within the gut precursor cells in a *Dronc* mutant background. Validating this result and our LOF experiments using the newly created *Dronc* alleles, we observed that genetic mosaics of a *Dronc*^{I29} null allele are larger than controls (Supplementary Figure 3). This data has been incorporated and referenced in the new version of the manuscript.

5. Figure 3. The authors generate a GFP/Myc-tagged *Dronc* construct under Actin promoter control. Given that this construct is driven under the ubiquitous Actin promoter, why is it only detectable in EBs? The fact that an exogenously expressed protein is expressed in EBs does not mean that the endogenous gene is expressed there, too. In fact, looking at the *Dronc*-Gal4 expression in sup Fig 5A, it appears to this author that the expression is largely in ECs.

Ultimately, the molecular reason why our Dronc-myc tagged protein is accumulated in EBs remains elusive. However, we want to highlight that this preferential accumulation is only detectable in conditions without tissue regeneration (Fig 3). Upon tissue damage, the construct is also accumulated in other cell types such as ECs (Supplementary Fig 6). Therefore, the accumulation of our chimeric protein appears to persist in Dronc-activating cells. This is also supported by our findings using the Dronc-activity sensor DBS-S-QF (Fig 3). We can speculate about how a non-active version of Dronc could be stabilised in Dronc-activating cells, but this is outside of the scope of the manuscript and it could compromise the narrative of the text. Therefore, we have not extended our explanation on this front. Considering that the results of our functional experiments correlate with the accumulation and activation of Dronc in EBs, it is conceivable to think that Dronc is expressed in EBs. Indeed, the widespread expression of *Dronc* was noticed by the reviewer in Supplementary Fig 6. Therefore, the accumulation of Dronc and its function in EBs cannot be linked to the transcriptional regulation of the gene. Together this data supports our hypothesis that Dronc is specifically required in EBs and this function is likely to be linked with post-transcriptional regulatory mechanisms.

6. Figure 4. The authors perform epistasis experiments with Dronc and Notch. It would be nice if they could confirm these assays with a Notch marker.

We would like to thank the reviewer for their suggestion since this has been one of the key findings added to the manuscript. The new version of the manuscript incorporates a analysis the well-established transcriptional reporter of Notch activity and EB identity marker, *Su(H)*-LacZ (3-5) (Figure 4). We observe that this reporter gene is broadly expressed in Dronc mutant conditions throughout the gut in different cell types, including fully differentiated ECs. Importantly, this up-regulation of *Su(H)*-LacZ occurs in flies reared in our experimental regime, without epithelial turnover. Validating the crosstalk between Dronc and Notch pathway, *Su(H)*-LacZ expression is rapidly downregulated upon differentiation under regenerative conditions (paraquat). This data consolidates our genetic interactions and indicates that Notch signalling is negatively regulated by Dronc in EBs. This fundamental result has prominently been highlighted and discussed in the text.

7. Regarding the control of Notch by Dronc, is there any genetic interaction between Dronc and Numb in the midgut. Do Numb mutants have a phenotype in the midgut?

We would like to thank the reviewer for their idea. Whilst we cannot completely rule out a regulatory role for Numb in combination with Dronc within the intestinal system, it is difficult to reconcile this possibility with our current data. Previous literature has previously shown that the functional interaction between Numb and Dronc in neuroblasts, relies exclusively on protein-protein interactions and it is fully independent of the enzymatic activity of Dronc. Indeed, Numb does not appear to contain Dronc cleavage sites. In our study, we are able to demonstrate that a proteolytically inactive version of Dronc, with a single amino acid substitution (C318A, Supplementary figure 4), recapitulated the *Dronc* LOF allele experiments in Figure 2. This implies a requirement for the catalytic activity of the protein rather than its sole presence, in order to regulate EB function. This suggests that Numb in this context has a minor contribution to our *Dronc* LOF phenotypes. Based on this, we consider this experiment unnecessary, although this possibility has been discussed in the discussion section of the manuscript.

8. In sup fig 3J and 3K, the authors express a C318A E352A mutant of Dronc. Due to the less penetrant phenotypes, the authors conclude that Dronc requires catalytic activity (blocked by C318A) for the non-apoptotic role in the midgut. However, this conclusion is not exact. E352A blocks the processing of Dronc into large and small subunits which might be the reason for the less penetrant phenotypes. Not sure why the authors did this experiment in an E352A background. It is also not clear why in sup fig 3K, they deleted the CARD domain.

We thank the reviewer for their query, understand their reasoning, and have sought to address their concerns. To achieve this, we generated a new allele which is a proteolytically inactive however remains cleavable (C318A). In this experimental condition we were able to recapitulate the findings of the C318A E352A *Dronc* allele, confirming a requirement for the catalytic activity of Dronc to regulate EB function. As a protein recruitment domain, the rationale for deleting the CARD domain was to establish if it had any effect on the regulation of EBs within this context. The data suggests however, that at least in combination with a (C318A E352A), its role is negligible.

Referee 2:

In this manuscript, the authors describe a non-apoptotic function of the initiator caspase Dronc in limiting the differentiation of enteroblasts under normal homeostasis. By using a number of novel genetic reagents, they establish the requirement of Dronc catalytic activity in enteroblasts, and not intestinal stem cells, to limit their differentiation into enterocytes. These findings are interesting as they further support the non-apoptotic functions of caspases and describes a cell-type specific function in the adult posterior gut in *Drosophila*

The authors suggest that the catalytic activity of Dronc is required rather than Dronc acting as a scaffold for protein interactions. This raises several question including how is Dronc activated and how is apoptosis prevented in the present of Dronc activity? While Dronc can undergo autoactivation, the requirement of Dark for Dronc activation should be assessed. On the basis of the DBS-S-QF sensor, the abundance and activation of Drice should also be assessed. Are there conditions where enteroblast differentiation can be blocked by Dronc overexpression or does this promote apoptosis? These issues need to be address and discussed in further detail.

In the current format, the findings are preliminary and require a more extensive evaluation that in the most part uses the reagents and methods they already have established. This should include a thorough analysis with appropriate controls of all of the cell types (e.g. *esg*, *delta*, *Su(H)*, *Pros* and *Pdm*) as well as markers of proliferation and cell death, for each genotype examined. The quantitation analysis should be kept consistent across the experiments. In addition, the rational for several experiments and the presentation order of the figures as well as the inclusion of sup data in the main figures could all be improved.

We would like to thank the reviewer for their comments and their general suggestions on how to improve the manuscript, we however disagree with them that the data is preliminary, since all of the information included is solid and supports the conclusions extracted from the data. Regardless we have tried to address many of their concerns and have attempted to incorporate additional key experiments in the manuscript that solidify our initial conclusions. Unfortunately, not all of the experiments suggested by the reviewer have been incorporated, since we consider some of them redundant or unnecessary.

- The reviewer raises an interesting question as to how Dronc is able to function without resulting in cell death. This remains an unanswered question within the field. A number of studies have suggested that the levels of Dronc activation are tightly controlled with a low to mild activation resulting in its non-apoptotic function, and high levels of activation leading to cell death. Whilst we openly state in the manuscript that we do not understand how Dronc is activated in the gut, our functional experiments probe its requirement in preventing EB differentiation in non-regenerative conditions. This is not incompatible with an alternative apoptotic role in the same cell when Dronc is highly activated. Indeed, this apoptotic role is likely to be required in tissue damaging conditions when an excess of intestinal precursors are generated (6). A paragraph in the discussion section has been incorporated describing all these possibilities in detail.

- The new version of the manuscript incorporates a dual labelling of the intestine combining DBS-S-QF and CasExpress as requested by the reviewer. As shown in the figure CasExpress is preferentially labelling differentiated ECs instead of EBs. This indicates that DBS-S-QF in non-tissue-damaging conditions mainly labels a non-apoptotic function of initiator caspases in EBs. Confirming this conclusion, all of the functional experiments removing different members of the apoptotic cascade apart from Dronc have failed to trigger differentiation phenotypes. This strongly supports the hypothesis that Dronc prevents EBs differentiating, acting in a non-apoptotic manner. All of the new experiments have been incorporated into the text and appropriately discussed in the different sections of the manuscript.
- In this new version of the text, we established that Dronc appears to act independently of Dark through the overexpression of two independent RNAis. We also demonstrate that upstream activators of the cascade are not involved through overexpression of a miRNA against hid, grim and reaper.
- We have attempted to overexpress Dronc within our tissues in order to observe its effects, however unfortunately this resulted in apoptosis. Therefore, the experiment suggested by the reviewer cannot be performed.
- We have confirmed the presence of proliferation in figure 2 by incorporating EDU experimental data into the text (Supplementary Figure 3).
- We are unsure as to which specific controls the reviewer feels the manuscript is lacking. We do however believe that including experimental data such as TUNEL staining and symmetrical cell identity makers for every experiment is excessive and will detract from the readability of the manuscript, increasing its complexity unnecessarily. We have however improved the presentation of the figures and the refurbished the text in line with the reviewers wishes. Additional experiments have been incorporated in different sections of text that support our initial hypothesis.

Referee #3:

In this paper, Arthurton et al. present data which they believe shows a function for the Caspase, Dronc, in enteroblast (EB) differentiation in the fly intestine. A non-apoptotic, differentiation function for a caspase in gut homeostasis is potentially quite interesting. The fly strains generated here, which have conditional alleles of Dronc, are elegant tools that should be generally useful for studying caspase function in diverse contexts. However, this paper has numerous technical flaws and shortcomings that led us to doubt its central conclusions. In addition there are many things that are not clearly presented. Specific comments follow below:

1. The introduction (lines 80-90) does not give an accurate or comprehensive summary of what the published literature reports about the mechanisms of the ISC-EB-EC differentiation pathway. For instance, well known roles for Esg, Pdm1, Stat, and su(H) are not mentioned. Since this paper is about EB differentiation, the introduction to this topic should be accurate.

We would like to thank the reviewer for their suggestion, and we agree with them. We have added a few sentences in the introductory section to better cover the topic suggested by the reviewer, but space limitations impede further explanations.

2. Using the REDDM cell lineage tracing system, the authors make the interesting and important point that under their conditions, ISCs in the midgut can be essentially quiescent. However there are many papers that show how rates of ISC division change as a function of gut region, fly age, sex, and mating. The authors need to specify clearly what gut region, sex, and age of flies they've used for their experiments.

We would like to thank the reviewer for this comment. Indeed, stating these points clearly within the text is important for the manuscript and therefore we have included these improvements. Additionally, to directly answer the reviewer's question we have incorporated low magnification images for a number of experiments. These images confirm a functional requirement for Dronc in EBs differentiation throughout the posterior midgut. New data and text clarifications have been added to the main manuscript to rectify concerns raised by the reviewer and improve the flow of the text.

3. In figures 1A and 1D, it looks like that the authors only show the R5 region of the midgut. Based on these pictures, it is hard to tell that DBS-S-QF is strikingly upregulated upon paraquat treatment. Much of the upregulation most likely happened in the malpighian tubules. The authors need to show whole midgut pictures for figures 1A and 1D. It could also help to quantify the DBS-S-QF fluorescence values.

We thank the reviewer for their suggestion. We have updated the picture of the DBS-S-QF to a more representative image. We have also included low magnification images of the DBS-S-QF sensor and the ReDDM tool within Figure 1. We have elected not to quantify fluorescence values since we have previously published quantifications of this nature in our previous manuscript (1) and the paraquat treatments consistently decorate a large proportion of the gut with the DBS-S-QF signal.

4. QF, which is activated in one of their conditional K/O strains, is known to be toxic for *Drosophila* cells. Thus the experiments with the QF-tagged Dronc allele may be misleading. The authors should

either test this, or use the Suntag allele for all experiments (it is already used for most experiments).

We thank the reviewer for this comment, we are indeed aware of the previously published toxicity related to QF in some circumstances. Indeed, the *suntag-HA-Cherry* allele was intentionally created to circumvent such a problem. To discard potential deleterious effects of *suntag-HA-Cherry*, additional constructs have been created containing this tag appended to *Dronc*. Importantly, the rescue construct containing *Dronc-suntag-HA-Cherry* functionally rescues a *Dronc* KO mutant background. Despite the previously reported toxicity of the QF protein, our alleles behave almost identically within the contexts previously tested. Altogether, this confirms the suitability of our alleles and solidifies our conclusions. Finally, we have obtained further validation of our alleles generating genetic mosaics of a previously null allele of *Dronc*; *Dronc*^{l29}.

5. Using the P35 overexpression data (Fig. 1G-1I) to suggest a non-apoptotic function of caspase in the gut is quite indirect and not so informative. It might be better to test the rate of gut regeneration in homozygous *Df(3L)H99* (*reaper/hid/grim* triple mutants) flies. Further, knockdown of *Dronc* in EBs in the *H99*^{-/-} background (apoptosis-deficient condition) should be more direct way to determine the non-apoptotic function of *Dronc* during EB-to-EC differentiation. The EB specific driver, *su(H)-Gal4-TS* can be used here. It is surprising this tool is not used in this manuscript.

We thank the reviewer for their comments. In the manuscript the overexpression of P35 forms only a part of our argumentation to suggest a non-apoptotic role for *Dronc*. In addition to P35, we also demonstrate that the overexpression of RNAi against all known effector caspases fails to replicate the *Dronc* LOF phenotypes, as does the knockdown of upstream regulators such as *Dark*. Furthermore, the elimination of the expression of pro-apoptotic factors using a microRNA against *Hid*, *Grim* and *reaper* (this experiment is equivalent to the one suggested by the reviewer using the *DfH99* but has the advantage of being able to simultaneously target the expression of these proteins in all gut precursors) does not mimic the *Dronc* mutant differentiation phenotypes. All of these factors strongly suggest that *Dronc* function in EBs is largely independent of the apoptotic pathway (and cell death) in our experimental conditions. In line of this argument, we also failed to observe any turnover of cells in the first seven days of *ReDDM* activation under our experimental regime. Since observations from our previous manuscript demonstrated the presence of historical and non-apoptotic apical caspase activity within the posterior midgut precursors (1), we conclude that there is a non-apoptotic role for *Dronc* in regulating EB proliferation and differentiation in non-regenerative conditions. Of course, this is not incompatible with an alternative apoptotic role in the same cells when *Dronc* is potentially activated under regenerative conditions. Indeed, this function is likely to emerge in tissue damaging conditions and basal epithelial turnover when an excess of gut precursors is generated (6). A paragraph in the discussion section has been incorporated describing in detail all these possibilities.

Please see our answer regarding the usage of *Su(H)-gal4* below, in point 7.

6. In Fig 1, the authors should plot the frequencies of cells that are Red+ Green+, Red+ Green-, and Red- Green+. This would be more informative.

We would like to thank the reviewer for their idea, however We feel that displaying the data in this way will be more confusing for the reader. Either way, the information described by the reviewer

can already be extracted from our graphs as they are currently presented. Furthermore, we would also like to clarify that owing to the design of the ReDDM tool, it is impossible to achieve cells which are Red- and Green+. This is because both the UAS-GFP and UAS-HistoneRFP are under the regulation of the *esg* promoter and are initially expressed at the same time. The GFP signal disappears when the Gal4 expression ceases, but one cannot switch off the RFP signal while keeping the GFP.

7. In Figure 2, the authors should use the Su(H)GBE-Gal4, a well-defined EB-specific driver, to knock out Dronc. The results might also be validated using standard FRT or MARCM clones mutant for Dronc, using another allele. In addition, the figure is not clear. What is the full genotype of the samples shown in 2A, 2B? Where is Suntag-Cherry? Counts of mitoses could also be useful here (to confirm Fig 2C).

We thank the reviewer for their suggestions. We elected not to use the Su(H)-Gal4 driver as previously published literature has suggested it is not specific to EBs, and it could include a subset of pre-EEs cells (7). This could lead to the generation of misleading results. We also consider that the experiment is redundant with our current mapping of the functional requirement of Dronc within intestinal precursors. We have completed *Dronc* LOF experiments specifically targeting its expression in the ISCs with *DI*-Gal4. This experiment showed that Dronc is not required in this cell type to prevent proliferation and differentiation in non-regenerative conditions. Conversely, the experiments using the *esg*-Gal4 driver have uncovered the novel functions of Dronc. Since the expression pattern of *esg*-Gal4 incorporates the expression of *DI*-Gal4 (within ISCs) and all known EB drivers, we hope the referee would agree with us that it is fair to conclude that there is a requirement for Dronc specifically in EBs. In addition, we have noticed the upregulation of the Notch-signalling pathway in Dronc mutant conditions, using *Su(H)*-lacZ as a reporter. Indeed, this marker remains active in fully differentiated ECs. This opens up the possibility of generating genetic stress in ECs using *Su(H)*-Gal4, due to the sustained production of flipase recombinase. This could lead to potentially generating inadvertent effects difficult to interpret.

The manuscript now incorporates mitotic clones within the midgut using the *Dronc*¹²⁹ null allele (Supplementary Figure 3). This experiment shows an increased clone size when compared to *Ry+* clones (Supplementary Figure 3), thus validating the results obtained with our alleles and confirming the effects of Dronc in progenitor cells.

The full genotypes for all experiments can be found in the materials and methods section, however the full genotype for 2A is *w*¹¹¹⁸; *esg*-Gal4 UAS-CD8-GFP : TubG80^{ts} UAS-Histone-RFP *Dronc*^{KO} / SM6A-TM6B and for 2B is *w*¹¹¹⁸; *esg*-Gal4 UAS-CD8-GFP / + ; TubG80^{ts} UAS-Histone-RFP *Dronc*^{KO} / UAS-FLP FRT *Dronc*-GFP-APEX FRT suntag-HA-Cherry. The suntag-HA-Cherry is expressed following excision of the wildtype cassette, however, is not visible using immunofluorescence techniques, likely due to the low levels of Dronc expression.

We agree with the reviewer and have included data for an EDU experiment in Supplementary Figure 3 to confirm the mitoses in Figure 2C.

All new data has been added to the manuscript as well as the corresponding text explanations.

8. The data in Fig 2 consistent with hyperplasia and proliferation could be explained by a requirement for Dronc in cell viability, rather than differentiation. The authors have not done

enough to rule out this possibility.

Please see response to point number 5.

9. The expression analysis of *Dronc* is poor. Endogenous *Dronc* or tagged active *Dronc* could be detected for some un-explained reason. An inactive tagged for was detected (Fig 3), but it is not clear that this mirrors the actual expression of endogenous *Dronc*. Moreover, the double staining in Fig 3 suggests the *Dronc* may be expressed in ISCs as well as EBs. The legend to this figure is not clear about what is shown.

We thank the reviewer for their comments but disagree to some extent with their statement. Unfortunately, the levels of *Dronc* expression are extremely low and not surprisingly, below the threshold of detection through standard immunohistochemistry in many cell types. This has been confirmed through our inability to detect different tagged versions of the *Dronc* protein expressed at endogenous levels in the gut and other tissues. Indeed, it is often necessary to use systems to amplify the signal to detect either *Dronc* as protein or its activity (8). Figure 4 of the manuscript shows the preferential accumulation of *Dronc* in EBs in non-regenerative conditions. This observation is confirmed by the colocalization of *Su(H)*-lacZ staining and the *Dronc* protein within EBs. These findings correlate with a specific functional requirement for *Dronc* in EBs, so it is conceivable that the EB specificity relies on the upregulation and activation of *Dronc* through unknown mechanisms in these cells. Following the reviewers' recommendation, we have improved the clarity of these descriptions and the corresponding figure legend.

10. The epistasis tests shown in Figure 4 are not meaningful. These merely suggest that Notch signaling and mTOR are dominant to *Dronc* in regulating differentiation. These results don't really imply that *Dronc* acts "as a negative regulator of the terminal differentiation program", as the authors suggest. The authors at least need some biochemistry or molecular data to support that *Dronc* acts through Notch or mTOR pathway to regulate EB-to-EC differentiation. For instance, examining the expression levels of the critical components of each pathway upon *Dronc* LOF or GOF could support their thesis. As noted above (point #8), many of the results presented here are also consistent with the possibility that loss of *Dronc* promotes gut epithelial turnover because *Dronc* is somehow essential for cell viability.

We thank the reviewer for their comment. In this regard we have included further analysis confirming the hypothesis that *Dronc* acts a negative regulator of Notch signalling (Figure 4). In our new set of experiments, we have incorporated the transcriptional reporter of Notch signalling and highly specific marker of EBs; *Su(H)*-LacZ. In *Dronc* KO intestines we observe increased expression of this Notch reporter in both EBs and differentiated cells, suggesting an increase in Notch activity. This strongly supports our initial hypothesis based on classical genetic epistasis that *Dronc* is required to negatively regulate Notch signalling in EBs. Supporting further this hypothesis, we have also noticed that *Su(H)*-lacZ is rapidly downregulated in differentiated ECs in regenerative conditions (Figure 4). Finally, this key experiment also suggests that ECs originating from *Dronc* mutant EBs are unable to effectively regulate key signalling pathways and are thus defective. This data also correlates with the qPCR data from Supplementary Figure 3, demonstrating an aberrant expression of EC metabolic proteins in intestines mutant for *Dronc*. Collectively this information validates our genetic epistasis

without the need for additional biochemical analysis, since the ultimate molecular link between Notch and Dronc remains unknown. This is currently also out of the scope of this work and it would require further experiments in the future.

REFERENCES

1. Baena-Lopez LA, Arthurton L, Bischoff M, Vincent JP, Alexandre C, McGregor R. Novel initiator caspase reporters uncover previously unknown features of caspase-activating cells. *Development*. 2018;145(23).
2. Xu D, Li Y, Arcaro M, Lackey M, Bergmann A. The CARD-carrying caspase Dronc is essential for most, but not all, developmental cell death in *Drosophila*. *Development*. 2005;132(9):2125-34.
3. Furriols M, Bray S. A model Notch response element detects Suppressor of Hairless-dependent molecular switch. *Curr Biol*. 2001;11(1):60-4.
4. Micchelli CA, Perrimon N. Evidence that stem cells reside in the adult *Drosophila* midgut epithelium. *Nature*. 2006;439(7075):475-9.
5. Ohlstein B, Spradling A. Multipotent *Drosophila* intestinal stem cells specify daughter cell fates by differential notch signaling. *Science*. 2007;315(5814):988-92.
6. Reiff T, Antonello ZA, Ballesta-Illan E, Mira L, Sala S, Navarro M, et al. Notch and EGFR regulate apoptosis in progenitor cells to ensure gut homeostasis in *Drosophila*. *EMBO J*. 2019;38(21):e101346.
7. Beehler-Evans R, Micchelli CA. Generation of enteroendocrine cell diversity in midgut stem cell lineages. *Development*. 2015;142(4):654-64.
8. Shinoda N, Hanawa N, Chihara T, Koto A, Miura M. Dronc-independent basal executioner caspase activity sustains *Drosophila* imaginal tissue growth. *Proc Natl Acad Sci U S A*. 2019;116(41):20539-44.

Dear Dr. Baena Lopez

Thank you for the re-submission of your revised manuscript to EMBO reports. I am truly sorry for the delay in getting back to you but we have only recently received the full set of referee reports and I have also asked the referees for further feedback.

As you will see, referee 2 now supports publication of the manuscript. However, while both referees 1 and 3 clearly appreciate that the manuscript has been improved, they are not convinced that the current data conclusively supports the conclusions made and we can therefore not offer publication of the manuscript in its current form. Referee 1 and 3 both consider experiments with the more specific Su(H)GBE-Gal4 driver essential to substantiate the conclusions, a notion that referee 1 emphasized again during further discussion. Referee 1 is concerned that the current approach to visualize Dronc expression is not convincing and that another assay should be employed. You have already informed me that you have data based on a Dronc-TurboID allele, which should ameliorate this concern. The stability of catalytically inactive Dronc needs to be verified. It should be clarified why ISC-specific loss of Dronc function does not affect the differentiation of the descendant cells, i.e., EBs. Importantly, referee 3 remains unconvinced that the current data is sufficient to conclude that Dronc restrains EB differentiation and alternative explanations need to be tested, as outlined. The data on InR signaling, can be removed, as you already suggested. The use of a stronger ISC driver is not essential, if the data convincingly show that DI-Gal4 has an 80% excision efficiency.

Given the constructive and supportive comments, I would like to give you the opportunity to address the remaining concerns in a second round of revision. Please note that - given the current situation - we have extended the revision time under our scooping protection to the time required to perform the necessary experiments. I suggest to keep in contact and to discuss the revision and timing further once more information on the duration of the current lockdown is available. We will reset the revision clock once experiments can be resumed.

- 1) Your manuscript has currently five figures and will be published as Report. This requires that you combine the Results and Discussion section. If the revision results in more than 5 figures your manuscript will be published as Article and the sections can stay separate.
- 2) I attach to this email a related manuscript file with comments by our data editors. Please address all comments and upload a revised file with tracked changes with your final manuscript submission.
- 3) Please provide up to five keywords.
- 4) Please add a callout to Figure 1D and Figure 2A, F, and G in the text, where appropriate.
- 5) We replaced Supplementary Information with Expanded View (EV) Figures and Tables that are collapsible/expandable online. A maximum of 5 EV Figures can be typeset. EV Figures should be cited as 'Figure EV1, Figure EV2" etc... in the text and their respective legends should be included in the main text after the legends of regular figures.

- For the figures that you do NOT wish to display as Expanded View figures, they should be

bundled together with their legends in a single PDF file called *Appendix*, which should start with a short Table of Content including page numbers. Appendix figures should be referred to in the main text as: "Appendix Figure S1, Appendix Figure S2" etc. See detailed instructions regarding expanded view here:

6) Our routine figure check that we perform on all revised manuscript indicated that in Figure 3C the green signal of the magnified image is much stronger than the green signal in the overview. We generally recommend to use as little contrast modification as possible. This applies to all fluorescent images.

7) We would also encourage you to include the source data for figure panels that show essential data. Numerical data should be provided as individual .xls or .csv files (including a tab describing the data). For blots or microscopy, uncropped images should be submitted (using a zip archive if multiple images need to be supplied for one panel). Additional information on source data and instruction on how to label the files are available .

8) Figure 1 is currently in landscape orientation, please change to portrait.

9) The following images lack scale bars. Please add them and define their size in the figure legend.

Missing scale bars in 2F-G, 3A, S2C, S6

Missing magnification scale bars in 2A, 3, 4D-I, S1A, S3, S6

10) Finally, EMBO reports papers are accompanied online by A) a short (1-2 sentences) summary of the findings and their significance, B) 2-3 bullet points highlighting key results and C) a synopsis image that is 550x200-400 pixels large (width x height) in .png format. You can either show a model or key data in the synopsis image. Please note that the size is rather small and that text needs to be readable at the final size. Please send us this information along with the revised manuscript.

Please let me know if you have any further questions. I am looking forward to receiving a revised version of your manuscript.

Kind regards,
Martin
Martina Rembold, PhD
Editor
EMBO reports

Referee #1:

I still have some major issues with this paper.

1. I still find the approach in Figure 3 troublesome. Expressing a gene under a heterologous

promoter to learn something about its normal expression is just not legitimate. On top of that, they are expressing the catalytic mutant which may have a completely different behavior compared to wild-type. It is too bad that the tagging of the endogenous gene doesn't work. But in my mind, this figure is seriously flawed.

2. The authors make a strong point that the function of Dronc in EBs is dependent on its catalytic activity, because the catalytic mutant has the same phenotype as the null mutant. However, an important control is missing here. We don't know if the catalytic mutant is a stable protein.

3. Along these lines, I asked the authors to consider that Dronc regulates Numb in this context. The authors dismissed this thought and didn't address this concern.

4. Another point which confuses me is the data in Figure 2F-I. Here, the authors target Dronc in ISCs using Df-Gal4. They say they target Dronc in more than 80% of the cells, but don't see the same phenotype they see with *esg*-Gal4 (Figure 2A-E). What confuses me is that when they target Dronc in ISCs, then the daughter cells of ISCs, including EBs, should also be mutant for Dronc. So, if Dronc has this important function in EBs, why don't they see the phenotype when Dronc is targeted in ISCs? The authors did not perform this experiment with an EB-specific (*Su(H)*-Gal4) driver. I find the answer to point 7 of reviewer 3 very unsatisfying that the *Su(H)*-Gal4 driver is also expressed in a subset of pre-EE cells. As the authors stated, the EEs are not affected by Dronc deficiency, so leaky expression in some pre-EEs will not affect the result.

5. While this paper was in revision, another paper appeared (Reiff et al., 2019; reference 57). Reiff et al. showed that expression of p35 increases the number of EBs in the posterior midgut, in contrast to what is shown in this paper. While the authors cite Reiff et al., this was done in passing and does not address this contrary finding. The authors need to address these opposite results.

6. The authors mention in line 200, that they used validated RNAi lines of the effector caspases in *Drosophila* and cite Leulier et al (reference 45). However, Leulier et al. only validated the RNAi line of *Dcp1*, but not the other caspases. The authors also didn't reveal which RNAi lines they were using in this manuscript, as there are many out there now. The same applies to dark RNAi.

7. Lines 130 and 345 contain typos. Line 232: Fig. 2L is 2M.

Referee #2:

I read with interest the comments of the other two Reviewers and the rebuttal letter from the author. As far as addressing my queries are concerned, they have extended their analysis where feasible and provided an improved manuscript. All reviewers had concerns in regard to verifying the non-apoptotic role for Dronc and the authors have provided further support for this by examining additional apoptotic regulators and reporters.

Referee #3:

The revision of this paper is improved in many respects, but still leaves some rather big questions open. Most importantly, we were still not convinced that Dronc's role in EBs is to specifically suppress premature differentiation into ECs, as opposed to other possible functions, for instance to restrain EB division or promote EB/EC viability. In essence, although the data presented are intriguing, we are not convinced by the authors' conclusions, depicted in the model (Fig 5b). Specifically:

1. It is quite odd that the authors refuse to do experiments we suggested in point 7, using the EB-specific Gal4 driver, su(H)GBE-Gal4. They cite Micchelli's paper and argue that the Su(H)GBE-Gal4 is not a specific EB driver, and could be expressed in a subset of pre-EEs. But this argument is non-sensical. A more recent paper published by Steven Hou clearly showed that Su(H)GBE-Gal4 is never expressed in the EE lineage, whereas esg-Gal4 (used extensively here) is. Testing the function of Dronc loss specifically in EBs is the most direct way to test their model that Dronc function in EBs accounts for these phenotypes, so they should try it. In this respect, it would also be useful for the authors to try the alternative ISC driver, esg-Gal4 su(H)-Gal80 (ts), which is much stronger than the DI-Gal4 driver. Much of the authors' argument rests on the lack of phenotype from Dronc K/O by DI-Gal4, but DI-Gal4 is notoriously weak and might not be giving an interpretable effect.

2. Regarding to the model in Fig 5, although Insulin-TOR pathway can act downstream of Notch signaling based on the previous literature, it is nevertheless inappropriate to include it in the model unless it has been thoroughly tested. Fig S7 has only a single result relevant to InR signaling, and it is not sufficient. At the least, differentiation markers should be scored and an InR gain of function rescue experiment should be presented.

In summary, the data presented here are very interesting, and the new tools for Dronc studies are stellar. But we were just not convinced that Dronc in EBs specifically restrains their differentiation. More data should be shown to support this novel idea. We wonder why "pre-mature differentiation" of EBs would give so much ISC division. We also wonder if Dronc K/O kills or delaminates EBs or new ECs non-apoptotically, thus activating regeneration and ISC division indirectly. We still wonder whether Dronc K/O in EBs might make them (not the ISCs) divide, through some interesting mechanism. These alternative explanations for the Dronc K/O phenotype can all be tested. If they were, the model could be presented with more confidence.

RESPONSE TO REVIEWERS

We want to thank the reviewers for their appreciation of our data and their constructive criticisms, particularly their reiterated interest in knowing the effects of eliminating Dronc expression exclusively in EBs. The associated results of this experiment have shown novel data that have improved our understanding of how Dronc works in non-apoptotic conditions, whilst expanding our working model. Of course, as always in science, it has opened new questions that deserve to be analysed in future experiments beyond the scope of this manuscript. Beyond the additional experimental information the new version of the manuscript has also improved the layout of the figures, text descriptions, and logic flow of the text. In the following lines, you can find a summary of the new experimental data and modifications incorporated into the manuscript.

- The new version of the manuscript incorporates pictures illustrating the preferential accumulation of Dronc in EBs using an endogenously tagged form of Dronc with TurboID.
- We provide an immunoprecipitation experiment demonstrating that the phenotypes attributed to the catalytically dead forms of Dronc are not a consequence of protein instability but the functional deficiencies of these mutants.
- We provide Gain- and Loss-of-function experiments showing that Numb is highly unlikely to be correlated with the Dronc phenotypes.
- We have specifically eliminated the expression of Dronc in EBs using the Su(H)-Gal4 driver. The experiment has shown an unexpected result, that makes our interpretation of how Dronc is functioning in non-apoptotic situations more complete. This experiment has shown that Dronc insufficiency in EBs is not able to generate hyperplasia and penetrant phenotypes equivalent to those obtained using esg-Gal4. However the EBs increase in size, likely as an indication of entering into the differentiation pathway. These results suggest that Dronc is required in both types of progenitor cells, but having a functionally active version in one of them is sufficient to preserve the progenitor cells in a non-proliferative and undifferentiated state. In this scenario, Dronc appears to ensure the coordinated behaviour of ISCs and EBs, thus preventing the reactivation of progenitor cells without demand.
- We agree with reviewer number 3 that the data referred to the with regards to the insulin pathway is tangential to the main argumentation of the manuscript, and does not bring additional information, therefore we have decided to remove it from the final version of the manuscript. This simplifies the manuscript, improving its clarity and the description of the results.
- The previous Figure number 5 corresponding to the model has been also removed from the manuscript and it will become the graphic summary of the manuscript.
- The text and figures have been amended to incorporate new experimental data, results, description and discussion. Important changes have been introduced in the text of the manuscript in the results and discussion section to this end.

Below you can find a detailed response to your comments (text in red).

Referee #1:

I still have some major issues with this paper.

1. I still find the approach in Figure 3 troublesome. Expressing a gene under a heterologous promoter to learn something about its normal expression is just not legitimate. On top of that, they are expressing the catalytic mutant which may have a completely different behavior compared to wild-type. It is too bad that the tagging of the endogenous gene doesn't work. But in my mind, this figure is seriously flawed.

We now show the expression of Dronc endogenously tagged with the biotin ligase TurboID (Shinoda, N., et al., *Dronc-independent basal executioner caspase activity sustains Drosophila imaginal tissue growth*. Proc Natl Acad Sci U S A, 2019. **116**(41): p. 20539-20544). This construct is able to report the presence of Dronc even though is physiologically expressed at very low levels, by transferring biotin to all those proteins in close proximity (Shinoda, N., et al., 2019). As shown in the new panel of Figure 4A, a preferential protein biotinylation is detected in EBs under non-regenerative conditions. This result confirms and validates our previous observations using our construct.

2. The authors make a strong point that the function of Dronc in EBs is dependent on its catalytic activity, because the catalytic mutant has the same phenotype as the null mutant. However, an important control is missing here. We don't know if the catalytic mutant is a stable protein.

We provide an immunoprecipitation followed by WB in Extended View Figure 2M. This WB shows that the stability of the FL-WT form of Dronc, the FLCAEA (catalytically dead), and the same construct with a truncated card domain (deltaCAEA) have comparable protein stability. This discards the correlation of our phenotypes with the lack of expression and stability of our constructs. On the contrary, our conclusions regarding the enzymatic requirement of Dronc.

3. Along these lines, I asked the authors to consider that Dronc regulates Numb in this context. The authors dismissed this thought and didn't address this concern.

We have performed the experiment of either reducing or overexpressing *numb* with the *esg*-Gal driver; however, none of these genetic manipulations caused noticeable phenotypes. This result and the published data indicating the non-enzymatic relation of the Numb-Dronc phenotypes in neuroblasts, strongly argues against any relevant implication of this factor in the intestinal Dronc-dependent phenotypes. For completeness, this information has been included in the current version of the manuscript as Appendix Figure 3 F-G.

4. Another point which confuses me is the data in Figure 2F-I. Here, the authors target Dronc in ISCs using DI-Gal4. They say they target Dronc in more than 80% of the cells, but don't see the same phenotype they see with *esg*-Gal4 (Figure 2A-E). What confuses me is that when they target Dronc in ISCs, then the daughter cells of ISCs, including EBs, should also be mutant for Dronc. So, if Dronc has this important function in EBs, why don't they see the phenotype when Dronc is targeted in ISCs? The authors did not perform this experiment with an EB-specific (Su(H)-Gal4) driver. I find the answer to point 7 of reviewer 3 very unsatisfying that the Su(H)-Gal4 driver is also expressed in a subset of pre-EE cells. As the authors stated, the EEs are not affected by Dronc deficiency, so leaky expression in some pre-EEs will not affect the result.

As we described in the initial figure 1 of the manuscript, our experiments have been performed in non-regenerative conditions, therefore although the ISCs become mutant, they do not differentiate as there is no demand for it. They remain in quiescence state. Supporting this, we have provided the ReDDM lineage tracing under our experimental conditions (Figure 1). Complementarily, we have shown that the modulation of multiple apoptotic regulators (e.g. P35, elimination of pro-apoptotic factors and effectors caspases) fail to induce the appearance of phenotypes (Extended view Figure 4).

We thank reviewer 1 and 3 for reiterating the need to perform the experiment with Su(H)-Gal4, since it has provided valuable information that has helped us to gain a greater understanding of how Dronc could work in non-regenerative conditions, whilst expanding our working model. Initially, we had some scientific concerns regarding the scientific potential of this experiment and technical difficulties (we found a genetic incompatibility between the driver and Dronc that impeded us from building the relevant stocks to set up the experiment) that made us reluctant to attempt it. After solving this problem by genetically cleaning the Gal4 driver, we managed to build the required stocks. The experiment demonstrated an unexpected result that in turn, has helped us to know better how Dronc is acting in non-apoptotic situations. Dronc deficiency in EBs increases their size but fails to cause hyperplasia and penetrant differentiation phenotypes equivalent to that obtained using *esg*-Gal4 driver (Figure 3). The combination of the results obtained with all of the Gal4 drivers suggests that Dronc activation is required in both progenitor cells but having it in one of them is sufficient to preserve them in quiescence (non-proliferative and undifferentiated). This model incidentally correlates very well with the observations using our Dronc tools that indicate the preferential accumulation/activation of Dronc in EBs, but not exclusively, since low levels are also detected in ISCs. In this scenario, Dronc appears to modulate not only the cell size of EBs, but also coordinates the activity of ISCs and EBs in order to retain them in a quiescent state. The ultimate molecular mechanism underlying these effects is unknown and beyond the scope of this manuscript, since it could be related to the regulation of specific proteins expressed in both cells (e.g. Sox21), the length-strength-timing of cell-cell interactions between ISCs and EBs, the regulation of specific intracellular pathways (e.g. Notch), the stress response upon exposure to specific environmental factors (e.g. microbiota interaction) etc... All of them are interesting questions to follow up in future work. These possibilities, the new information and the model have been included and discussed in the current version of the manuscript.

5. While this paper was in revision, another paper appeared (Reiff et al., 2019; reference 57). Reiff et al. showed that expression of p35 increases the number of EBs in the posterior midgut, in contrast to what is shown in this paper. While the authors cite Reiff et al., this was done in passing and does not address this contrary finding. The authors need to address these opposite results.

The overexpression of either one or two copies of P35 does not cause noticeable phenotypes in our experimental regime. Comparable results were also observed expressing RNAi lines against effector caspases. Therefore the disparities with the manuscript cited by the reviewer could possibly emerge from the status of the epithelia in our experimental setting (with negligible cell turnover over) and the manuscript cited (with minimal but still present cell turn over). Taking into consideration this point, we have discussed the manuscript cited by the reviewer in the section of the discussion that we have considered better aligned with the experimental results.

6. The authors mention in line 200, that they used validated RNAi lines of the effector caspases in *Drosophila* and cite Leulier et al (reference 45). However, Leulier et al. only validated the RNAi line of Dcp1, but not the other caspases. The authors also didn't reveal which RNAi lines they were using in this manuscript, as there are many out there now. The same applies to dark RNAi.

We apologise for the lack of information regarding some of the lines used in our study, this has been included in the current version of the manuscript.

Regarding the specific comment of the RNAi lines of effector caspases, the manuscript of Leulier, F and collaborators (Leulier, F, et al., *Systematic in vivo RNAi analysis of putative components of the Drosophila cell death machinery*. *Cell Death Differ*, 2006. **13**(10): p. 1663-74) provides information about the lines used in our study. In Figure 3 for the reviewer (see below), we show how different combinations of RNAis can only reduce apoptosis after irradiation (since their functional redundancy has been reported previously), whereas the concomitant downregulation of all these genes abolishes apoptotic cell death (TUNEL staining). This information has not been included in the manuscript since the characterisation of our RNAi lines has been done before and have been used extensively in the literature.

Figure 3 for the reviewer shows the TUNEL immunostaining in wing imaginal of different genetic backgrounds in 8 hrs. after irradiation. Combinations of effector caspases in pairs under the regulation of *enGal4* reduced the level of TUNEL in the posterior compartment (left panel), while all of them in combination completely abolishes apoptosis (right panel).

7. Lines 130 and 345 contain typos. Line 232: Fig. 2L is 2M.

The order of figures has changed to accommodate the new data

Referee #2:

I read with interest the comments of the other two Reviewers and the rebuttal letter from the author. As far as addressing my queries are concerned, they have extended their analysis where feasible and provided an improved manuscript. All reviewers had concerns in regard to verifying the non-apoptotic role for Dronc and the authors have provided further support for this by examining additional apoptotic regulators and reporters.

Referee #3:

The revision of this paper is improved in many respects, but still leaves some rather big questions open. Most importantly, we were still not convinced that Dronc's role in EBs is to specifically suppress premature differentiation into ECs, as opposed to other possible functions, for instance to restrain EB division or promote EB/EC viability. In essence, although the data presented are intriguing, we are not convinced by the authors' conclusions, depicted in the model (Fig 5b). Specifically:

1. It is quite odd that the authors refuse to do experiments we suggested in point 7, using the EB-specific Gal4 driver, *su(H)GBE-Gal4*. They cite Micchelli's paper and argue that the *Su(H)GBE-Gal4* is not a specific EB driver, and could be expressed in a subset of pre- EEs. But this argument is non-sensical. A more recent paper published by Steven Hou clearly showed that *Su(H)GBE-Gal4* is never expressed in the EE lineage, whereas *esg-Gal4* (used extensively here) is. Testing the function of Dronc loss specifically in EBs is the most direct way

to test their model that *Dronc* function in EBs accounts for these phenotypes, so they should try it. In this respect, it would also be useful for the authors to try the alternative ISC driver, *esg-Gal4 su(H)-Gal80 (ts)*, which is much stronger than the *DI-Gal4* driver. Much of the authors' argument rests on the lack of phenotype from *Dronc* K/O by *DI-Gal4*, but *DI-Gal4* is notoriously weak and might not be giving an interpretable effect.

We want to clarify that we agree with the reviewer that the *DI-Gal4* is a weak driver and this is an important problem if one wants to express UAS transgenes in a consistent manner. However, this concern is not applicable to our experiments since we show that 80% of the *DI+* cells successfully excised the FRT rescue cassette and therefore become permanently mutant from that moment onwards. Despite removing the expression of *Dronc* in the majority of *DI*-expressing cells, we did not observe phenotypic manifestations.

We thank reviewer 1 and 3 for reiterating the need to perform the experiment with *Su(H)-Gal4*, since it has provided valuable information that has helped us to gain a greater understanding of how *Dronc* could work in non-regenerative conditions, whilst expanding our working model. Initially, we had some scientific concerns regarding the scientific potential of this experiment and technical difficulties (we found a genetic incompatibility between the driver and *Dronc* that impeded us from building the relevant stocks to set up the experiment) that made us reluctant to attempt it. After solving this problem by genetically cleaning the *Gal4* driver, we managed to build the required stocks. The experiment demonstrated an unexpected result that in turn, has helped us to know better how *Dronc* is acting in non-apoptotic situations. *Dronc* deficiency in EBs increases their size but fails to cause hyperplasia and penetrant differentiation phenotypes equivalent to that obtained using *esg-Gal4* driver (Figure 3). The combination of the results obtained with all of the *Gal4* drivers suggests that *Dronc* activation is required in both progenitor cells but having it in one of them is sufficient to preserve them in quiescence (non-proliferative and undifferentiated). This model incidentally correlates very well with the observations using our *Dronc* tools that indicate the preferential accumulation/activation of *Dronc* in EBs, but not exclusively, since low levels are also detected in ISCs. In this scenario, *Dronc* appears to modulate not only the cell size of EBs, but also coordinates the activity of ISCs and EBs in order to retain them in a quiescent state. The ultimate molecular mechanism underlying these effects is unknown and beyond the scope of this manuscript, since it could be related to the regulation of specific proteins expressed in both cells (e.g. *Sox21*), the length-strength-timing of cell-cell interactions between ISCs and EBs, the regulation of specific intracellular pathways (e.g. Notch), the stress response upon exposure to specific environmental factors (e.g. microbiota interaction) etc... All of them are interesting questions to follow up in future work. These possibilities, the new information and the model have been included and discussed in the current version of the manuscript.

2. Regarding to the model in Fig 5, although Insulin-TOR pathway can act downstream of Notch signaling based on the previous literature, it is nevertheless inappropriate to include it in the model unless it has been thoroughly tested. Fig S7 has only a single result relevant to InR signaling, and it is not sufficient. At the least, differentiation markers should be scored and an InR gain of function rescue experiment should be presented.

In order to simplify the manuscript and the description of the results, we have eliminated all the information related to the Insulin pathway from the manuscript.

In summary, the data presented here are very interesting, and the new tools for *Dronc* studies are stellar. But we were just not convinced that *Dronc* in EBs specifically restrains their differentiation. More data should be shown to support this novel idea. We wonder why "pre-mature differentiation" of EBs would give so much ISC division. We also wonder if *Dronc* K/O kills or delaminates EBs or new ECs non-apoptotically, thus activating regeneration and ISC division indirectly. We still wonder whether *Dronc* K/O in EBs might make them (not the ISCs) divide, through some interesting mechanism. These alternative explanations for the *Dronc* K/O phenotype can all be tested. If they were, the model could be presented with more confidence.

Dear Alberto,

Thank you for the submission of your revised manuscript to EMBO reports. We have now received the reports from referee 1 and 3 (copied below).

As you will see, both referees find that the study has been significantly improved during revision and recommend publication after minor revisions to text and figures. It will be sufficient to discuss how Dronc might regulate Notch signaling, no further experimental data are required from our side. Please correct Figure 3 as discussed.

From the editorial side, there are also a number of things that we need before we can proceed with the official acceptance of your study:

1) Your manuscript has currently 5 figures and will be published in our Reports section. For this, we need you to combine the Results and Discussion section. Please also keep an eye on character count (25,000 +/- 2,000 characters for the main text)

2) Please arrange the manuscript sections in the order listed in our guide to authors. See also <https://www.embopress.org/page/journal/14693178/authorguide#textformat>

3) Data availability section: Please add a statement saying "This study includes no data deposited in external repositories"

4) Please shorten the title to 100 characters incl spaces

5) Please add callouts to Fig 2A,F+G in the text where appropriate

6) Please provide scale bars in magnifications of Fig 2A,b; 4A,E,F; 5D-I; App Fig S2A,H+O; App Fig S3D.

7) I attach here the synopsis image at its final size (550 pixels width). I think that the resolution and text size could be improved a bit.

8) Appendix:

- Please provide a detailed table of content including page numbers

- add the 'S' to the label on the figures themselves (Appendix Figure S1 etc).

- Legends: please define the sizes of all scale bars in the legend and make sure that the description of all graphs showing quantifications contains this information

N = x, (biological or technical replicate), the nature of the error bar is defined, the statistical test used is defined, the meaning of e.g. ** is defined.

- Fig S3D: please add scale bars for the inserts

- Fig S3F, G: please define the meaning of the white stippled lines and complete the information on UAS-Numb-RNAi (BL31.....)

9) Our production/data editors have asked you to clarify several points in the figure legends (see attached document). Please incorporate these changes into your final manuscript text and return the revised file with tracked changes with your final manuscript submission

With kind regards,

Martina

Martina Rembold, PhD
Editor
EMBO reports

Referee #1:

This manuscript has been revised a few times and I am now satisfied with it. Overall, it is nice work and it defines a new non-apoptotic of the caspase Dronc in Drosophila. I am glad to see that our suggestion to test Su(H)-Gal4 in their assays helped to further refine the model.

However, I have one formal point. There is an inaccuracy in Figure 1G. The authors place Dark in the wrong location in the pathway. Dark is not upstream of Dronc and is not directly activated by Hid/Rpr/Grim. Instead, Hid/Rpr/Grim inactivate DIAP-1 as the authors describe in the Introduction which releases Dronc which can then form the apoptosome with Dark. The authors need to correct this model.

There is one typo in line 332.

Referee #3:

In this revision, Arthurton et al. have finally added the experiments we requested using the EB-specific Su(H)-Gal4Ts driver. Contrary to expectations, the depletion of Dronc in EBs failed to induce significant gut hyperplasia and/or differentiation phenotypes. These results, unfortunately, weaken the general conclusions regarding the functions of Dronc in EBs. Precisely how Dronc regulates ISC proliferation and EB differentiation is less clear than it appeared to be. The results with Dronc-RNAi expressed using ISC and EB drivers separately (Line 228-247) are somewhat confusing, as neither recapitulated the phenotypes given using esg-Gal4, which expresses in both ISCs and EBs. To the authors' credit, these new findings are clearly described. Nevertheless, despite these rather disappointing new results, the non-apoptotic phenotype of Dronc in the gut is clear, and quite impressive. Moreover, the Dronc-related tools generated in this paper will very useful for the apoptosis field. Non-apoptotic functions of caspases are an area of great interest currently, and this paper makes a contribution. Given this, we would support this paper for publication in EMBO Reports, even though the mechanistic function of non-apoptotic Dronc in the gut stem cell lineage is not 100% clear. Several specific comments are listed below:

1. The introduction should summarize published information on non-apoptotic caspase functions, especially in stem cells. This information is highly important to set the context of this study. Other topics that are covered in detail in the Introduction, for instance about signaling in the Drosophila midgut, can be made more concise.

2. The paper has many small grammatical mistakes (often with articles), and should be proofed by a native speaker.
3. Line 144: please explain how the DBS-S-QF caspase activity reporter works.
4. In Fig. 3E-F, it would be better to include Pdm1 staining.
5. Although the genetic data locate Dronc upstream of the Notch-pathway, precisely how Dronc regulates Notch signaling is still unknown. To add more mechanistic insight here, we suggest the authors examine whether Dronc regulates the expression levels of the Notch protein in EBs. Furthermore, exploring other critical downstream target genes of Notch signaling in addition to the Su(H)GBE-lacZ reporter could be informative.

Referee #1:

This manuscript has been revised a few times and I am now satisfied with it. Overall, it is nice work and it defines a new non-apoptotic of the caspase Dronc in *Drosophila*. I am glad to see that our suggestion to test Su(H)-Gal4 in their assays helped to further refine the model.

However, I have one formal point. There is an inaccuracy in Figure 1G. The authors place Dark in the wrong location in the pathway. Dark is not upstream of Dronc and is not directly activated by Hid/Rpr/Grim. Instead, Hid/Rpr/Grim inactivate DIAP-1 as the authors describe in the Introduction which releases Dronc which can then form the apoptosome with Dark. The authors need to correct this model.

The figures has been amended accordingly.

There is one typo in line 332.

This has been corrected.

Referee #3:

In this revision, Arthurton et al. have finally added the experiments we requested using the EB-specific Su(H)-Gal4Ts driver. Contrary to expectations, the depletion of Dronc in EBs failed to induce significant gut hyperplasia and/or differentiation phenotypes. These results, unfortunately, weaken the general conclusions regarding the functions of Dronc in EBs. Precisely how Dronc regulates ISC proliferation and EB differentiation is less clear than it appeared to be. The results with Dronc-RNAi expressed using ISC and EB drivers separately (Line 228-247) are somewhat confusing, as neither recapitulated the phenotypes given using *esg-Gal4*, which expresses in both ISCs and EBs. To the authors' credit, these new findings are clearly described. Nevertheless, despite these rather disappointing new results, the non-apoptotic phenotype of Dronc in the gut is clear, and quite impressive. Moreover, the Dronc-related tools generated in this paper will very useful for the apoptosis field. Non-apoptotic functions of caspases are an area of great interest currently, and this paper makes a contribution. Given this, we would support this paper for publication in EMBO Reports, even though the mechanistic function of non-apoptotic Dronc in the gut stem cell lineage is not 100% clear. Several specific comments are listed below:

1. The introduction should summarize published information on non-apoptotic caspase functions, especially in stem cells. This information is highly important to set the context of this study. Other topics that are covered in detail in the Introduction, for instance about signaling in the *Drosophila* midgut, can be made more concise.

The introduction has been made shorter but we have added the info requested regarding the background.

2. The paper has many small grammatical mistakes (often with articles), and should be proofed by a native speaker.

The leading of the manuscript is native speaker and the manuscript has carefully been revised by him

3. Line 144: please explain how the DBS-S-QF caspase activity reporter works.

This info has been added in the main text.

4. In Fig. 3E-F, it would be better to include Pdm1 staining.

This was a mistake in the description of this information. The text has been amended to add the appropriate information.

5. Although the genetic data locate Dronc upstream of the Notch-pathway, precisely how Dronc regulates Notch signaling is still unknown. To add more mechanistic insight here, we suggest the authors examine whether Dronc regulates the expression levels of the Notch protein in EBs. Furthermore, exploring other critical downstream target genes of Notch signaling in addition to the Su(H)GBE-lacZ reporter could be informative.

This has been discussed in the discussion section of the manuscript.

Prof. Luis Alberto Baena Lopez
University of Oxford
Sir William Dunn School of Pathology
South Parks Road
Oxford UK
Oxford, State... Ox13RE
United Kingdom

Dear Alberto,

I am very pleased to accept your manuscript for publication in the next available issue of EMBO reports. Thank you for your contribution to our journal.

At the end of this email I include important information about how to proceed. Please ensure that you take the time to read the information and complete and return the necessary forms to allow us to publish your manuscript as quickly as possible.

As part of the EMBO publication's Transparent Editorial Process, EMBO reports publishes online a Review Process File to accompany accepted manuscripts. As you are aware, this File will be published in conjunction with your paper and will include the referee reports, your point-by-point response and all pertinent correspondence relating to the manuscript.

If you do NOT want this File to be published, please inform the editorial office within 2 days, if you have not done so already, otherwise the File will be published by default [contact: emboreports@embo.org]. If you do opt out, the Review Process File link will point to the following statement: "No Review Process File is available with this article, as the authors have chosen not to make the review process public in this case."

Should you be planning a Press Release on your article, please get in contact with emboreports@wiley.com as early as possible, in order to coordinate publication and release dates.

Thank you again for your contribution to EMBO reports and congratulations on a successful publication. Please consider us again in the future for your most exciting work.

Kind regards,
Martina

Martina Rembold, PhD
Editor
EMBO reports

THINGS TO DO NOW:

You will receive proofs by e-mail approximately 2-3 weeks after all relevant files have been sent to our Production Office; you should return your corrections within 2 days of receiving the proofs.

Please inform us if there is likely to be any difficulty in reaching you at the above address at that time. Failure to meet our deadlines may result in a delay of publication, or publication without your corrections.

All further communications concerning your paper should quote reference number EMBOR-2019-48892V4 and be addressed to emboreports@wiley.com.

Should you be planning a Press Release on your article, please get in contact with emboreports@wiley.com as early as possible, in order to coordinate publication and release dates.

Corresponding Author Name: Luis Alberto Baena Lopez

Journal Submitted to: EMBO reports

Manuscript Number: EMBOR-2019-48892V2-Q